# From cyclic ice streaming towards Heinrich-like events: the grow-and-surge instability in the Parallel Ice Sheet Model

Johannes Feldmann[1] and Anders Levermann[1,2,3]

[1]Potsdam Institute for Climate Impact Research (PIK), Potsdam, Germany
[2]Institute of Physics, University of Potsdam, Potsdam, Germany
[3]LDEO, Columbia University, New York, USA

*Correspondence to:* Johannes Feldmann (johannes.feldmann@pik-potsdam.de)

**Abstract.** Here we report on a cyclic, physical ice-discharge instability in the Parallel Ice Sheet Model, simulating the flow of a three-dimensional, inherently buttressed ice-sheet-shelf system which periodically surges on a millennial timescale. The thermo-mechanically coupled model on 1 km horizontal resolution includes an enthalpy-based formulation of the thermo-dynamics, a non-linear stress-balanced based sliding law and a very simple sub-glacial hydrology. The simulated unforced surging is characterized by rapid ice streaming through a bed trough, resulting in abrupt discharge of ice across the grounding line which is eventually calved into the ocean. We visualize the central feedbacks that dominate the sub-sequent phases of ice build-up, surge and stabilization which emerge from the interaction between ice dynamics, thermodynamics and the sub-glacial till layer. Results from the variation of surface mass balance and basal roughness suggest that ice sheets of medium thickness may be more susceptible to surging than relatively thin or thick ones for which the surge feedback loop is damped. We also investigate the influence of different basal sliding laws (ranging from purely plastic to non-linear to linear) on possible surging. The presented mechanisms underlying our simulations of self-maintained, periodic ice growth and destabilization may play a role in large-scale ice-sheet surging, such as the surging of the Laurentide Ice Sheet, which is associated with Heinrich Events, and ice-stream shut-down and reactivation, such as observed in the Siple Coast region of West Antarctica.

## 1 Introduction

Glacial surging is characterized by rapid speed-up of ice flow and abrupt increase in ice discharge. Surging of glaciers found in the mountainous regions of the Earth is observed to occur on a decadal to centennial timescale, depending on regional conditions (e.g., Meier and Post, 1969; Jiskoot et al., 2000; Yasuda and Furuya, 2015; Sevestre and Benn, 2015; Herreid and Truffer, 2016). Repeated activation and stagnation of ice streams which drain the Siple Coast region of the West Antarctic Ice Sheet (e.g., Retzlaff and Bentley, 1993; Fahnestock et al., 2000) alters the flow pattern and mass balance of this part the ice sheet on a centennial time scale (Joughin and Alley, 2011; Kleman and Applegate, 2014). During glacial periods, quasi-periodic, large-scale surging of the Laurentide Ice Sheet likely led to massive iceberg calving into the ocean on a millennial time scale (MacAyeal, 1993; Clarke et al., 1999). These so-called Heinrich Events (Heinrich, 1988; Broecker et al., 1992; Kirby and Andrews, 1999) are associated with substantial freshening of the North Atlantic, reduction of the Atlantic meridional

overturning circulation (McManus et al., 2004) and are connected to abrupt climate changes on a global scale (Bond et al., 1993; Broecker, 1994; Hemming, 2004; Mohtadi et al., 2014).

Periodic oscillations of ice-sheet growth and surging have been suggested either to be of unforced type and thus originating from ice-internal mechanisms ("binge-purge" oscillations, see MacAyeal, 1993) or to be driven externally, i.e., by climate forcing (e.g., Hulbe, 2004; Alvarez-Solas et al., 2013; Bassis et al., 2017). Numerical modeling studies that investigated ice-sheet-intrinsic surging include the demonstration of creep instability (Clarke et al., 1977) and hydraulic runaway (Fowler and Johnson, 1995) as possible main feedbacks that drive unforced surging, the application to the Laurentide Ice Sheet to simulate its quasi-periodic surging (Marshall and Clarke, 1997; Calov et al., 2002; Greve et al., 2006; Papa et al., 2006; Calov et al., 2010; Roberts et al., 2016), the simulation of (cyclic) ice streaming and stagnation reminiscent of the flow variability of the Siple Coast ice streams (Alley, 1990; Pattyn, 1996; Payne and Dongelmans, 1997; Fowler and Schiavi, 1998; Bougamont et al., 2011; Van Pelt and Oerlemans, 2012; Robel et al., 2013) and the investigation of ice-stream oscillations in interaction with bed topography under the influence of ice-shelf buttressing (Robel et al., 2016). Model complexity ranges from the consideration of a simple slab of ice (e.g. Clarke et al., 1977) to the solution of the full Stokes equations to simulate real-world problems using satellite data (Kleiner and Humbert, 2014). Limitations to several of these studies include the restriction to the flow-line case (only one horizontal dimension considered), the prescription of a strongly idealized bed geometry (flat bed or inclined plane), and the use of simplified parameterizations of ice-internal, lateral and basal stresses (e.g., basal sliding chosen to be proportional to the driving stress).

Here we apply a channel-type bed geometry to a three-dimensional state-of-the-art ice sheet model to simulate the cyclic surging of a marine ice-sheet-shelf system. In particular, and in contrast to several previous studies, our simulations use a sliding law that is based on the stress balance of the ice and thereby has stress boundary conditions and that incorporates the effect of overburden pressure and basal melt water on the bed strength in non-linear fashion. In other words, the computed sliding velocity is not a direct parameterization through the local basal conditions but results from solving the non-local shallow-shelf approximation of the stress balance and at the same time, in combination with the basal properties, determines the basal stresses. Our model includes a minimal version of a subglacier, i.e., a basal till layer underlying the ice which interacts with the ice sheet through melt-water exchange; an interaction which is crucial to model unforced cyclic ice-sheet growth and surge. The nature of the chosen three-dimensional topographic setup allows to simulate complex ice flow and inherently emerging ice-shelf buttressing. Analyzing the modeled surge cycle, we identify competing fundamental mechanisms that underlie successive ice build-up, surge and stabilization. These mechanisms are visualized by the means of feedback loops. We also investigate conditions that lead to the damping of the oscillations in our model and explore how different sliding laws affect ice-flow characteristics. Eventually we discuss our results and conclude.

In a previous study, oscillatory surge behavior has been investigated with an earlier version of the model used here (Van Pelt and Oerlemans, 2012). In contrast to the marine ice-sheet-shelf system modeled here, they simulated a qualitatively very different type of an ice body, i.e., a land-terminating glacier. The main differences regarding the models (linear vs. non-linear friction law, transport of basal till water) as well as the experimental setups (bed entirely above sea level vs. marine bed trough, different boundary conditions) are detailed in the methods section and are picked up in the discussion section. Results from

Van Pelt and Oerlemans (2012) include the occurrence of high- and low-frequency oscillations as well as steady fast ice flow in the partially sampled parameter space, spanned by two sliding-law parameters, i.e., the till friction angle and the sliding-law exponent. The frequency and magnitude of the surge cycle was also found to be sensitive to variations in the climate forcing (surface temperature and mass balance).

## 2  Methods

### 2.1  Model

We use the open-source Parallel Ice Sheet Model (PISM; Bueler and Brown, 2009; Winkelmann et al., 2011; PISM authors, 2017), version stable07 (https://github.com/pism/pism/). The thermo-mechanically coupled model applies a superposition of the shallow-ice approximation (SIA; Morland, 1987) and the shallow-shelf approximation (SSA; Hutter, 1983) of the Stokes stress balance (Greve and Blatter, 2009). In particular, the SSA allows for stress transmission across the grounding line and thus accounts for the buttressing effect of laterally confined ice shelves on the upstream grounded regions (Gudmundsson et al., 2012; Fürst et al., 2016). The ice rheology is determined by Glen's flow law (Cuffey and Paterson, 2010). An energy-conserving enthalpy formulation of the thermodynamics in particular allows for an advanced calculation of the basal melt rate for polythermal ice (Aschwanden et al., 2012). The model applies a linear interpolation of the freely evolving grounding line and accordingly interpolated basal friction, and uses one-sided differences in the driving stress close to the grounding line (Feldmann et al., 2014).

A nonlinear Weertman-type sliding law is chosen to calculate the basal shear stress $\boldsymbol{\tau_b}$, based on the sliding velocity of the ice $\boldsymbol{u_b}$ with a sliding exponent $q = \frac{1}{3}$ as used in several previous studies (e.g., Schoof, 2007; Goldberg et al., 2009; Hindmarsh, 2009; Gudmundsson et al., 2012; Pattyn et al., 2013; Feldmann and Levermann, 2015b; Asay-Davis et al., 2016)

$$\boldsymbol{\tau_b} = -\tau_c \frac{\boldsymbol{u_b}}{u_0^q \, |\boldsymbol{u_b}|^{1-q}}. \tag{1}$$

Here $\tau_c$ is the till yield stress (Bueler and van Pelt, 2015), which evolves in time with changing basal till water content and overburden pressure (see below). For simplicity we set the velocity scaling parameter $u_0$ to $1\,\mathrm{m\,s^{-1}}$ (unit of the sliding velocity calculated in the model, as chosen in, e.g., Pattyn et al., 2013; Feldmann et al., 2014; Feldmann and Levermann, 2015a). Note that $\boldsymbol{u_b}$ results from solving the non-local SSA stress balance (Bueler and Brown, 2009, Eq. 17) in which $\boldsymbol{\tau_b}$ appears as one of the terms that balance the driving stress. This implementation of basal sliding is substantially different (and introduces more complexity) compared to several models that have previously been used in attempt to model cyclic surging, where $\boldsymbol{u_b}$ is a local function of $\boldsymbol{\tau_b}$, the latter often given by the negative of the driving stress at the ice base (e.g. Payne and Dongelmans, 1997; Fowler and Schiavi, 1998; Papa et al., 2006; Calov et al., 2002, 2010; Robel et al., 2013).

The till yield stress in Eq. (1) is determined by a Mohr-Coulomb model (Cuffey and Paterson, 2010)

$$30 \quad \tau_c = \tan(\phi)\, N_0 \left(\frac{\delta P_o}{N_0}\right)^s 10^{\frac{e_0}{C_c}(1-s)},\qquad(2)$$

that accounts for the effect of evolving ice thickness $H$, the associated change in overburden pressure $P_o = \rho_i g H$ on the basal till, and the amount of water stored in the till $W_{til}$. Here $s = W_{til}/W_{max}$ is the fraction of the water layer thickness in the till with respect to a fixed maximum layer thickness $W_{max}$ (Bueler and van Pelt, 2015). All other parameters are prescribed and are constant in space and time (adopted from Bueler and van Pelt, 2015, see Table 1 for a full list of parameters, their naming and values). Note that there is an upper bound to the yield stress enforced in the model, which is determined by the overburden pressure, i.e., $\tau_{c,\max} = \tan(\phi)\, P_o$ (for details see Bueler and van Pelt, 2015, Sec. 3.2). Compared to Van Pelt and Oerlemans (2012), where the yield stress is linear in the product of the overburden pressure and the till water fraction ($\tau_c \sim (1-s)P_o$; see their Eqs. 2 and 4), the yield-stress formula applied here is an exponential function of $s$ (Eq. 2). This allows for a non-linear evolution of the till yield stress during drainage or fill-up of the basal water layer.

The sub-glacial model is a slightly modified version of the undrained plastic bed model of (Tulaczyk et al., 2000a, b), as described in (Bueler and van Pelt, 2015, Section 3). The term undrained refers to the fact that this model does not account for horizontal transport of melt water stored in the basal till and thus melt water is produced and consumed only locally (in contrast to Van Pelt and Oerlemans (2012), where there is also horizontal diffusion of till water; see their Eq. 1). The evolution equation for the till-stored water thickness, $W_{til}$, is a function of the local basal melt rate $m$ (positive for melting, negative for refreezing)

$$\frac{\partial W_{til}}{\partial t} = \frac{m}{\rho_w} - C_d.\qquad(3)$$

The drainage-rate parameter $C_d$ allows for drainage of the till in the absence of water input. The water-layer thickness is bounded ($0 \leq W_{til} \leq W_{til}^{max}$) to avoid unreasonably strong filling of the till with melt water. Melt water which exceeds $W_{til}^{max}$ is not conserved.

## 2.2 Experimental setup

The three-dimensional setup is designed to model a marine ice sheet, which drains through a bed trough, feeding a bay-shaped ice shelf which calves into the ocean. The idealized bed topography (Fig. 1) is a superposition of two components: the bed component in x direction, $b_x(x) = -150\,\mathrm{m} - 0.84 \cdot 10^{-3}\,|x|$, is an inclined plane, sloping down towards the ocean (Fig. 1b). The component in y direction, $b_y(y)$, has channel-shaped form (Fig. 1c) and is a widened version of the one used in the MISMIP+ experiments (Asay-Davis et al., 2016, here with adjusted parameters for domain width and channel side-wall width, see Table 1). The superposition of both components yields a bed trough which is symmetric in both x and y directions (symmetry axes $x = 0$ and $y = 0$). While the main ice flow is in x direction (from the interior through the bed trough towards the ocean) there is also a flow component in y direction, i.e., from the channel's lateral ridges down into the trough. Resulting

convergent flow and associated horizontal shearing enable the emergence of ice-shelf buttressing, which leads to a grounding-line position further downstream than in the absence of an ice shelf. Ice is cutoff from the ice shelf and thus calved into the ocean beyond a fixed position (Fig. 1a). There exist more sophisticated methods to represent calving in a numerical model (e.g., Nick et al., 2010; Levermann et al., 2012; Pollard et al., 2015) leading to more realistic calving behavior and calving-front geometry. Our simple approach of prescribing a calving front (fixed in space and time) far enough from the region of the confined ice shelf makes sure that the calving front does not interact with grounding-line migration in the course of the surge cycle. Due to the symmetry of the setup we only consider the right-hand half of the domain throughout our analysis.

Compared to Van Pelt and Oerlemans (2012), who model a land-based glacier that rests on a bed entirely above sea level and exhibits an unconfined tongue-shaped (convex) ice-stream terminus (see their Fig. 2), here we simulate a marine ice-sheet-shelf system. This allows to model oscillatory flow of a topographically confined ice-stream which exhibits a concave grounding line, being buttressed by the downstream bay-shaped ice shelf (Fig. 1). Our model domain is larger by a factor of about 6 and 2 in the x and y directions, respectively, resulting in a 500 km long ice sheet, compared to the glacier of about 100 km length modeled in Van Pelt and Oerlemans (2012).

Surface mass balance, surface ice temperature and geothermal heat flux are assumed to be constant and spatially uniform (in contrast to Van Pelt and Oerlemans, 2012, where surface mass balance and temperature are parameterized by the bedrock elevation, prescribing an accumulation/ablation zone). There is no melting beneath the ice shelf. Glacial isostatic adjustment is not accounted for in the experiments. The simulations are initiated with a block of ice from which an ice-sheet-shelf system evolves while ice flow, basal mechanics and till-stored water content adjust. This spinup lasts a few 1000 yr and thus we focus on the time after this phase.

The model is run using finite differences and a regular grid of 1 km horizontal resolution. An initial examination of the flow field reveals that the SIA velocities are small compared to the SSA velocities in our simulation. Despite this fact, considering the SIA in the simulations in particular allows for the representation of a three-dimensional temperature field.

## 3 Results

### 3.1 Cyclic surging

For the given set of parameters (Table 1) the ice-sheet-shelf system takes on an oscillatory equilibrium of continuously alternating phases of surge and growth. This unforced behavior has been illustratively described as "binge-purge" mechanism by MacAyeal (1993) using a minimal model of ice (thermo-)dynamics and geometry (slab of ice, with thickness and temperature being spatially uniform along the horizontal axis Macayeal, 1993). This mechanism has been further investigated in later studies, including models of higher complexity in terms of ice dynamics as well as setup geometry (e.g., ice thickness, basal hydrology and other properties evolving non-uniformly in space and time; application to a three-dimensional, Laurentide-characteristic bed topography; coupling to a climate model; see for instance Calov et al., 2002; Van Pelt and Oerlemans, 2012; Robel et al., 2013; Roberts et al., 2016). Here we basically describe the same mechanism, focusing on the competing internal feedback mechanisms that dictate the "binge-purge" cycle and that affect the ice dynamics on different time scales (Fig. 2),

based on results from a sophisticated, three-dimensional ice sheet model. At the same time our results reveal the possibility of cyclic growth and surge of an inherently buttressed marine ice-sheet-shelf system.

On the slow time scale, the ice sheet tends to grow toward an equilibrium thickness, which is determined by the balance between snowfall and ice flux (velocity). If this equilibrium ice thickness is too large to be sustained by the basal conditions, this build-up (negative gray feedback loop in Fig. 2) is interrupted by an abrupt surge event with a rapid, self-enforcing speed-up of the ice flow (positive feedback loop in red). The associated large-scale ice discharge into the ocean eventually leads to a stabilization of the shrunken ice-sheet-shelf system (negative feedback loop in blue), which again tends to restore a balance thickness before a new surge event kicks in.

At the beginning of the modeled surge cycle the negative feedback loop of slowing-down ice growth is dominant: the basal till water content drops close to zero and basal friction is high, allowing gradual thickening of the ice sheet (Figs. 3, 4 and A2). The thickening causes an increase in basal melt water production due to the lowering of the pressure melting point at the ice base. The increasing water content in the basal till attenuates further increase in basal friction (which still increases due to the effect of ice thickening, i.e., growing overburden pressure $P_o$, see Eq. 2), leading to an increase in ice discharge and thus reducing further thickening. In the absence of any other mechanisms, the ice sheet would hence reach a steady state as ice thickening would approach zero, eventually.

However, the continuous accumulation of water in the sub-glacial till during the slow build-up initiates a surge event before the equilibrium thickness is reached. The self-enforcing feedback of rapid ice speed-up becomes dominant: lowered friction at the well lubricated ice-sheet base leads to an acceleration of ice flow through the bed trough (Fig. 3 and A1). In turn, this causes an increase in strain and frictional heating due to enhanced shearing inside the ice sheet and sliding of the ice over the bed, respectively (Fig. A2 and A3). The resulting additional melt water production further lubricates the ice base, leading to even more speed-up (termed "hydraulic runaway" by Fowler and Johnson, 1995). Inside the bed trough, the previously relatively stagnant ice flow has entered a state of rapid ice streaming (velocities at several $\mathrm{km\,yr^{-1}}$, Figs. 1a, b and 4d). The ice streaming is additionally fostered by the effect of strain heating at the side margins of the trough (Fig. 4): faster flow causes stronger shearing of the ice, resulting in more heat production which in turn softens the ice, allowing for more shearing and thus flow acceleration (so called "creep instability", Clarke et al., 1977, see positive feedback loop in our Fig. 5).

The ice streaming inside the bed trough leads to enhanced downstream advection from the ice sheet's thick interior into the ice shelf, manifesting a pronounced peak in ice discharge and iceberg calving, respectively (Figs. 3d and e). The associated damping feedback between ice velocity (ice flux) and thickness (blue stabilization loop in Fig. 2) counteracts the self-enforcing feedback between ice velocity and till water (red surge loop). On the long term, this discharge-related thinning of the ice sheet leads to the end of the surge as melt water production decreases, basal friction increases and ice flow decelerates. When the ice sheet has become too thin to maintain insulation of its base from the cold atmosphere then the basal melt rate drops. The associated decrease in till water now is amplified through the same mechanism that was responsible for the till water increase during the surge phase (red loop) and the ice stream shuts down. At some point basal refreezing sets in, consuming further water from the till layer (Fig. A2). As the water content in the drained till drops close to zero and thus bed friction quickly

increases, the ice sheet can build up again. The period duration of a whole surge cycle is of about 1800 yr, from which the slow build-up phase takes about 80 %.

35     We would like to note that the domain-averaged yield stresses resulting from our simulations (order of $\sim 100$ kPa, see Fig. 3c) are relatively large compared to values from *in situ* and laboratory experiments (order of $\sim 1$ kPa to $\sim 10$ kPa, see Table 7.5 in Cuffey and Paterson, 2010). In our experiments the highest values occur during the build-up phase which is when the water content in the till is very close to zero ($s \approx 0$) for which Eq. (2) yields $\tau_c(s = 0) \sim 10^5$ kPa. Though in the model $\tau_c$ is limited by the overburden pressure, the maximum possible value in our experiments is still on the order of $\sim 10^3$ kPa

(assuming an ice thickness of $1000$ m; for a visualization see Fig. 1 in Bueler and van Pelt, 2015). Such large values occur predominantly in the regions outside of the bed channel and in the thick interior of the ice sheet where the basal till layer is continuously dry and ice flow is stagnant, biasing the domain-average towards high values. In contrast, inside the lubricated bed channel the simulated yield stresses are much lower, especially during the phases of ice streaming (on the order of $\sim 10$ kPa), lying within the observatory range. This is in accordance with the fact that the observational values were inferred from till

samples stemming from regions of relatively fast ice(-stream) flow (e.g., Truffer et al., 2000; Tulaczyk et al., 2000a; Kamb, 2001).

## 3.2   Surge damping

Varying the bed strength in our simulations, we find that surging is maintained in a cyclic manner (oscillatory equilibrium) only if the bedrock roughness allows the evolution of an ice sheet of medium thickness. For rather slippery basal conditions, realized

by low values of the till friction angle, $\phi \leq 8\,^\circ$, and thus rather thin ice sheets, surging occurs initially but then is damped such that on the long term the ice sheet reaches a non-oscillating stable equilibrium state (Fig. 6). Decreasing the value of $\phi$ within this regime leads to faster damping and a shorter cycle duration. For sufficiently lubricated (thin) ice sheets no surging takes place at all. In contrast to the case of maintained cyclic surging, the ice flow enters a state of continuous streaming at velocities of several $100$ m yr$^{-1}$ with stable till water thickness (Figs. 6b and c).

Vice versa, increasing the friction angle towards large values yields rougher beds, promoting the evolution of thicker ice sheets which surge at larger magnitude. The surge frequency first increases (between $\phi = 10\,^\circ$ and $\phi = 30\,^\circ$), before decreasing again (Fig. 7). For sufficiently strong beds with $\phi \geq 60\,^\circ$ (and thus comparatively thick ice sheets) initial surging is damped, similarly to the case of relatively low values of $\phi$ discussed above. Consequently, surging is maintained only in a regime of medium bed strength (medium values of $\phi$) that promote ice sheets of medium thickness. Damped surging occurs on both ends

of this regime (above an upper and below a lower critical threshold of $\phi$), i.e., for relatively strong and weak beds.

    We investigate changes in the ice-flow characteristics close to the lower regime boundary in response to a small modification of the basal roughness. For this purpose we perturb the above equilibrium ice sheets of oscillatory ($\phi = 10\,^\circ$) and non-oscillatory ($\phi = 8\,^\circ$) type by decreasing/increasing the value of $\phi$. Our results show that when the friction is lowered from $\phi = 10\,^\circ$ to values of $\phi \leq 8\,^\circ$ then the originally surging ice sheet undergoes damping and enters a stable steady state, eventu-

ally (Fig. 8a). Hence, the flow characteristics of the perturbed ice sheet are more or less the same as in the spin up experiments when using the same values of the friction angle (compare Figs. 6a and 8). In contrast, perturbing the system in the other di-

rection, i.e. increasing the friction angle from $\phi = 8\,°$ (Fig. 8b) then maintained surging does only occur for values of $\phi \geq 20\,°$ (compared to $\phi = 10\,°$ for the case of ice-sheet spinup). For lower values of $\phi$ the ice flow starts to oscillate initially but then goes back into a state of stable flow at the same velocity as before, whereas now ice-sheet thickness and till water content are larger (both increasing for increasing $\phi$). Thus, the ice sheet in stable equilibrium requires a comparatively large perturbation of the basal conditions in order to turn into a state of maintained surging. In contrast, a small perturbation is sufficient to bring the continuously oscillating ice sheet into a stable steady state.

The finding from above, that thin ice bodies are less likely to surge than ice sheets of medium thickness, is supported by additional experiments with reduced surface accumulation $a$. According to these simulations, lower accumulation results in thinner ice sheets, a weaker surge amplitude and a longer surge-cycle duration (Fig. 9). The longer cycle duration can be explained by the fact that less snowfall causes the ice sheet to take longer to grow thick enough to trigger a surge event. At the same time the formation of basal till water during build-up takes longer and the kick-off of the surge event requires a smaller

amount of till water (and thus a thinner ice body). Below a threshold of a fifth of the default value ($a = 0.075\ \mathrm{myr}^{-1}$) a rather thin steady-state ice sheet forms and surging is not existent anymore.

## 3.3   Role of basal sliding law

The above results show cyclic or damped surging for a confined set of parameter values (default surface accumulation $a$ and till friction angle $\phi$ given in Table 1 are only slightly varied). These simulations use a particular non-linear sliding law, determined

by a basal sliding exponent of $q = \frac{1}{3}$ (Eq. 1). In general, this exponent can range from $q = 0$ (purely plastic sliding law) to $q = 1$ (linear sliding law). To explore the influence of the basal sliding law on the ice flow behavior, we conduct further simulations, sampling $q$ between 0 and 1 at an interval of $\frac{1}{12}$. For each applied parameter value of $q$ the till friction angle $\phi$ (Eq. 2) is varied between $5\,°$ and $85\,°$, spanning a wide range from relatively slippery to very rough bed conditions, respectively. The resulting $q - \phi$ parameter space is explored in terms of surge-cycle duration and ice-sheet volume (Fig. 10). Due to the large number

of simulations the experiments are carried out on a grid of $5\ \mathrm{km}$ horizontal resolution (in contrast to $1\ \mathrm{km}$ used in the default simulations).

The results show that (damped) surging occurs in a range from $q = \frac{1}{12}$ to $q = \frac{3}{4}$ (circles and triangles in Fig. 10). Within this regime larger values of $q$ correspond to higher friction angles $\phi$, i.e., going towards a more linear friction law requires a rougher bed in order to observe (damped) surging. Maintained surging occurs in a smaller range, i.e., from $q = \frac{1}{6}$ to $q = \frac{5}{12}$. This regime

is embedded such that the transition from the oscillatory state into the stable regime in most cases leads through the damped regime. Generally, decreasing $q$ or increasing $\phi$ yield a longer period duration of the surge cycle while the mean grounded ice mass increases. This can be explained by considering the relevant acting stresses: a larger value of $\phi$ leads to a larger magnitude of the basal yield stress (Eq. 2) and thus a stronger basal shear stress (Eq. 1). In the shallow-shelf approximation the driving stress (due to surface slope of the ice sheet) is balanced by a combination of the membrane stresses (responsible for ice-flow

acceleration) and the basal shear stresses (see Eq. 17 and the following paragraph in Bueler and Brown, 2009). An increase in basal shear thus slows down ice speed-up, promoting a longer period of ice-sheet build-up and larger ice-sheet thickness. Decreasing the exponent $q$ leads to the same results because also here the magnitude of the basal shear stress increases. This

becomes evident from Eq. (1) where the fraction $(|\boldsymbol{u_b}|/u_0)^q$ increases with decreasing $q$ since $|\boldsymbol{u_b}|/u_0 < 1$. The duration of the surge cycle ranges from about 1800 yr to 2700 yr for maintained surging (mean $\approx 2200$ yr) and from 800 yr to 5700 yr for damped surging (mean $\approx 2600$ yr).

For the particular cases of purely plastic ($q = 0$) or linear sliding ($q = 1$) no surging occurs in our simulations, which is independent of $\phi$. In the vicinity of $q = 0$ most of the experiments produce a stable and rather thick ice sheet (squares in Fig. 10), whereas around $q = 1$ the ice sheets become very thin (a few 10 m of thickness). In some cases these very thin ice sheets do not have any grounded ice inside the bed channel and thus lack comparability (marked by an "x" in Fig. 10). In general, very small values of $\phi$ cause continuous streaming of a rather thin ice sheet on a slippery bed, whereas large $\phi$ values lead to rough basal conditions allowing the evolution of a comparatively thick steady-state ice sheet (Fig. 7). Thus, only those ice sheets which are not too large or small show surging behavior. This confirms and generalizes our specific results from Sec. 3.2 that there is a thickness regime in which surging occurs whereas too thin or too thick ice sheets reach a stable equilibrium.

## 4    Discussion and conclusions

We model the cyclic surging of a three-dimensional, inherently buttressed, marine ice-sheet-shelf system (Fig. 1). Periodically alternating ice growth and surge are unforced and emerge from interactions between the dynamics of ice flow (evolution of velocity, internal and basal stresses, ice thickness), its thermodynamics (heat conduction, strain and basal frictional heating, melt-water production) and the subglacier (melt-water storage and drainage).

We identify three consecutive phases throughout the surge cycle (ice build-up, surge and stabilization), each characterized by a dominating feedback mechanism which we visualize in a feedback-loop scheme (Fig. 2). These feedbacks of slowing-down ice thickening, rapid ice speed-up and discharge, and decelerating ice thinning (Figs. 3 and A2) can explain central processes that likely prevailed during repeated large-scale surging of the Laurentide Ice Sheet and the associated Heinrich Events of global-scale impact. During the surge phase mainly the process of hydraulic runaway (positive feedback between basal melt water production and flow acceleration; Fowler and Johnson, 1995) is in effect. It is complemented by creep instability (positive feedback between strain heating and ice deformation; Clarke et al., 1977), which additionally promotes rapid ice streaming (Figs. 4 and 5). The modeled cyclic alternation of ice streaming and stagnation provides a simple example of ice-stream shut-down and re-activation, a phenomenon which is characteristic for the dynamics of some of the Siple Coast outlets in West Antarctica.

Our results suggest that medium-sized ice sheets are more susceptible to cyclic surging than rather thin or thick ones. We find a transition from surge to non-surge behavior (surge damping) of the ice flow when decreasing/increasing the thickness of the surging ice body in our simulations, realized by applying lower/larger basal roughness or surface mass balance (Figs. 6 and 9) or by a variation of the friction exponent in the sliding law (Fig. 10). This is consistent with the existence of a critical minimum ice thickness found by Schubert and Yuen (1982). According to their results, exceeding this thickness threshold enables the occurrence of creep instability, potentially leading to rapid surging. Furthermore, our results reveal that an ice

sheet in stable equilibrium requires a comparatively large perturbation of the basal conditions in order to turn into a state of maintained surging, whereas a small perturbation is sufficient to bring the continuously oscillating ice sheet into a stable steady state (Fig. 8).

Compared to the observed interval of about $7,000$ yr at which Heinrich Events re-occured during the last glacial period (Hemming, 2004), our modeled surge-cycle period of $\sim 2,000$ yr is much shorter. This is not surprising, given that our idealized model setup on a synthetic bed geometry is not designed and the parameters are not tuned to represent conditions that prevailed for the prehistoric Laurentide Ice Sheet. Thus, we refer to studies designed to model this ice sheet when it comes to the proper representation of the characteristic surge frequency of Heinrich Events (e.g., Marshall and Clarke, 1997; Calov et al., 2002; Papa et al., 2006; Roberts et al., 2016). Our model results are closer to results from conceptual studies which also use an idealized geometry (e.g., Bougamont et al., 2011; Van Pelt and Oerlemans, 2012; Robel et al., 2016). These studies all yield a surge-cycle duration of $\sim 1,000 - 2,000$ yr, despite considerable differences in degree of physical approximations, parameterizations and complexity in setup geometry. However, all of them use a Weertman-type, stress-balanced based sliding law (Eq. 1) and are based on the same (though individually modified) sub-glacial model (Tulaczyk et al., 2000a, b), suggesting that both have a strong imprint on the surge-cycle duration.

Conducting a parameter study that explores the $q-\phi$ space reveals that both decreasing the sliding exponent $q$ and increasing the friction angle $\phi$ lead to an increase of the surge-cycle duration (Fig. 10). The dependence of the cycle duration on $q$ is in accordance with results from (Van Pelt and Oerlemans, 2012), who used a previous version of PISM and a qualitatively different topographic setup (see Methods section and below). Sampling the parameter space between $q = 0$ and $q = 0.3$ they were able to model maintained oscillation also for the case of purely-plastic basal sliding ($q = 0$ in Eq. 1), which in our simulations only exists for $q$ ranging between $\frac{1}{6}$ and $\frac{5}{12}$. High-frequency oscillations with a period duration of $\sim 100$ yr as found in their experiments (in addition to the "low-frequency" cycle duration of $\sim 1000$ yr) are not present in our simulations.

Differences in the results compared to the PISM study of Van Pelt and Oerlemans (2012) are a combination of differences in 1) the model versions, 2) the experimental setups and 3) the choice of model parameters. The main difference between the two model versions is given by the applied friction law, i.e., being linearly vs. non-linearly determined by the overburden pressure and the amount of water in the till layer (see Methods section). One could expect that this substantial difference would reflect in the time scales of the modeled surge cycles. Though the absolute cycle durations differ significantly between the two studies, the relative timing of the surge initiation and the relative duration of the surge event (both with respect to the full cycle duration) are very similar (surge phase takes about $\sim 20\,\%$ of the full cycle).

The different experimental setups, in particular the prescribed bed topographies of different character, yield ice bodies of qualitatively very different characteristics, i.e, an unbuttressed, tongue-shaped, land-terminating glacier vs. an inherently buttressed marine ice-sheet-shelf system (the latter being much larger in horizontal extent). Buttressing increases the period duration of the surge cycle and decreases the magnitude of grounding-line migration as shown in (Robel et al., 2014). Consistently, we obtain a longer cycle duration (about a factor of 2) and a smaller magnitude of grounding-line migration (relative to the ice-sheet length) compared to Van Pelt and Oerlemans (2012). Maximum sliding velocities during the surge phase are on the same order of magnitude ($|\boldsymbol{u_b}| \sim 1000$ m/yr) but still substantially faster in our simulations. This is facilitated by the

complete saturation of the till layer in almost the entire bed trough and a temporary loss of buttressing due to the advance of

the central portion of the grounding line in the course of the surge (compare our Figs. 3, A1 and 4 to Figs. 6 and 7 in Van Pelt and Oerlemans, 2012). Our results confirm that complex boundary conditions (e.g., spatially varying surface temperature and accumulation, as applied in Van Pelt and Oerlemans, 2012) are not a requirement in order to obtain cyclic surging.

Note that for the conducted $q - \phi$ parameter study our choice of the velocity scaling parameter $u_0$ (Table 1) leads to a large spread of possible magnitudes of the basal shear stress, ranging from $|\tau_b| \sim 100$ kPa for $q = 0$ to $|\tau_b| \sim 1$ Pa for $q = 1$

(calculated for typical values of $\tau_c \sim 100$ kPa and $|u_b| \sim 1000$ m/yr). Choosing a parameter value on the order of magnitude of the sliding velocity (e.g., $u_0 \sim 1000$ m/yr, or $u_0 \sim 100$ m/yr, as done in Van Pelt and Oerlemans, 2012) would result in a much more confined $|\tau_b|$ value and thus facilitate the comparison of the results. However, scanning the $q - \phi$ parameter space with values of $u_0$ on these orders of magnitude revealed that in the given setup no surge type oscillations occur but a stable steady state ice sheet emerges. This confirms that the differences in the results between the two PISM studies can only be

attributed to the combined effect of model version, experimental setup and parameter values.

The surface accumulation is found to be a further parameter with strong influence on the surge-cycle duration in our simulations. Less snowfall leads to a longer duration of the surge cycle (Fig. 9) since the ice sheet takes longer to grow thick enough to trigger a surge event. This correlation between surface accumulation and surge frequency is also found in other studies modeling surge events (e.g., Greve et al., 2006; Calov et al., 2010). However, a decrease of the surge magnitude with decreasing

snowfall as found in our simulations is not present in these studies, whereas this behavior is consistent with results from the PISM study of Van Pelt and Oerlemans (2012) who find a decrease in surge amplitude when increasing the equilibrium line altitude. One essential difference between the corresponding applied models and PISM is that they parameterize the sliding velocity through the local basal conditions whereas in PISM the sliding velocity results from solving the non-local shallow-shelf approximation of the stress balance. Besides several other differences in model type and geometric setup, this might also be

the cause why a variation of basal sliding does not affect the period duration of a surge cycle in these studies, contrary to our findings.

Several other parameters in our model likely have an effect on the occurrence of surging and its dynamics (e.g., the overburden-pressure fraction $\delta$ in Eq. 2, the till drainage rate $C_d$ in Eq. 3, as well as surface temperature, geothermal heat flux and bed slope). However, further investigation of the parameter-dependency of the surging behavior (e.g., as done for sur-

face temperature and geothermal heat flux in Robel et al., 2014) is beyond the scope of this study. In fact, it aims at reporting on the realization of cyclic surging/ice-streaming of an ice-sheet-shelf system in the Parallel Ice Sheet Model based on suitable model components and justified set of parameters.

*Author contributions.* J.F. and A.L. designed research; J.F. performed research; J.F. and A.L. analyzed data and wrote the paper

*Competing interests.* The authors declare no conflict of interest.

*Acknowledgements.* The research leading to these results has received funding from the Deutsche Forschungsgemeinschaft (DFG) under priority program 1158 (LE 1448/8-1). Development of PISM is supported by NASA grants NNX13AM16G and NNX13AK27G. The authors also gratefully acknowledge the European Regional Development Fund (ERDF), the German Federal Ministry of Education and Research and the Land Brandenburg for providing resources on the high performance computer system at the Potsdam Institute for Climate Impact Research.

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

**Table 1.** Physical constants and model parameters

| Parameter | Value | Unit | Physical meaning |
|---|---|---|---|
| $n$ | 3 | | Exponent in Glen's flow law |
| $q$ | $\frac{1}{3}$ | | Basal friction exponent |
| $u_0$ | 1 | $\mathrm{m\,s^{-1}}$ | Scaling parameter for basal velocity in the sliding law |
| $\phi$ | 10 | ° | Till friction angle |
| $N_0$ | 1000 | Pa | Reference effective pressure (Bueler and van Pelt, 2015) |
| $\delta$ | 0.02 | | Parameter determining effective overburden pressure $\delta P_o$ (Bueler and van Pelt, 2015) |
| $e_0$ | 0.69 | | Reference void ratio at $N_0$ (Bueler and van Pelt, 2015) |
| $C_c$ | 0.12 | | Till compressibility (Bueler and van Pelt, 2015) |
| $W_{til}^{max}$ | 2 | m | Maximum water in till (Bueler and van Pelt, 2015) |
| $C_d$ | 0.001 | $\mathrm{m\,yr^{-1}}$ | Till drainage rate (Bueler and van Pelt, 2015) |
| $\rho_i$ | 918 | $\mathrm{kg\,m^{-3}}$ | Ice density |
| $\rho_w$ | 1000 | $\mathrm{kg\,m^{-3}}$ | Fresh-water density |
| $\rho_{sw}$ | 1028 | $\mathrm{kg\,m^{-3}}$ | Sea-water density |
| $g$ | 9.18 | $\mathrm{m\,s^{-2}}$ | Gravitational acceleration |
| $L_x$ | 700 | km | Length of right-hand half of the symmetric domain |
| $L_y$ | 160 | km | Width of domain (entering Eq. 4 of Asay-Davis et al., 2016) |
| $f_c$ | 16 | km | Characteristic width of channel side walls (entering Eq. 4 of Asay-Davis et al., 2016) |
| $d_c$ | 500 | m | Depth of bed trough compared with side walls (entering Eq. 4 of Asay-Davis et al., 2016) |
| $w_c$ | 24 | km | Half-width of bed trough (entering Eq. 4 of Asay-Davis et al., 2016) |
| $x_{cf}$ | 640 | km | Position of fixed calving front in right-hand half of domain |
| $a$ | 0.3 | $\mathrm{m\,yr^{-1}}$ | Surface accumulation rate |
| $G$ | 70 | $\mathrm{mW\,m^{-2}}$ | Geothermal heat flux |
| $T_s$ | -20 | °C | Surface temperature of the ice |

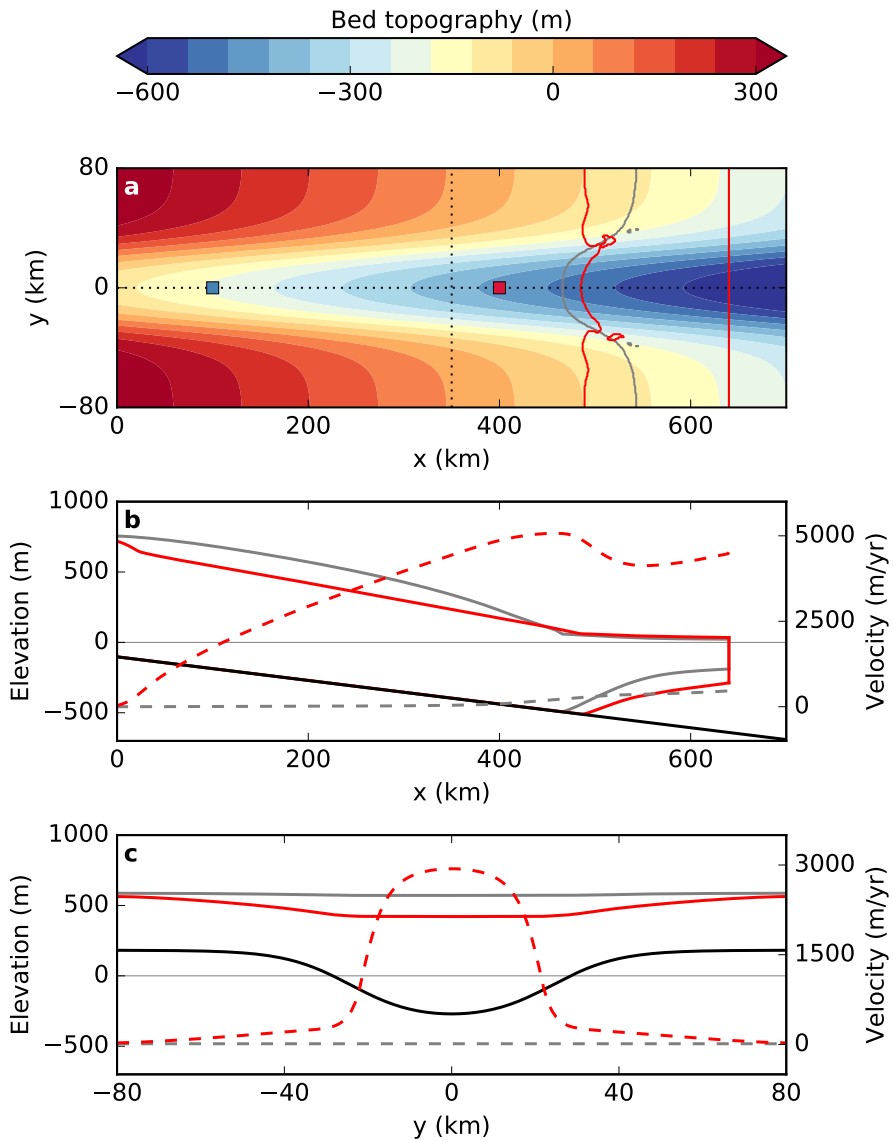

**Figure 1. (a)** Bed topography prescribed in the experiments (colorbar) with contours of grounding line and calving front during build-up (gray) and surge (red; see corresponding circles in Fig. 3). Note that throughout this study we focus on the right-hand-half of the symmetric model domain as shown here (symmetry axis at $x = 0$). Dotted lines mark locations of the cross sections shown in the other two panels. **(b)** Cross section in x direction along the centerline of the model domain. Profiles of the ice sheet (straight lines) and its velocity (dashed) are shown for the build-up phase (gray) and during surge (red), bed topography in black. **(c)** Cross section in y direction across the model domain at $x = 350$ km. Same colors as in panel (b). Red and blue squares on the centerline denote locations of two point measurements for which timeseries are shown in Figs. A1 and A3.

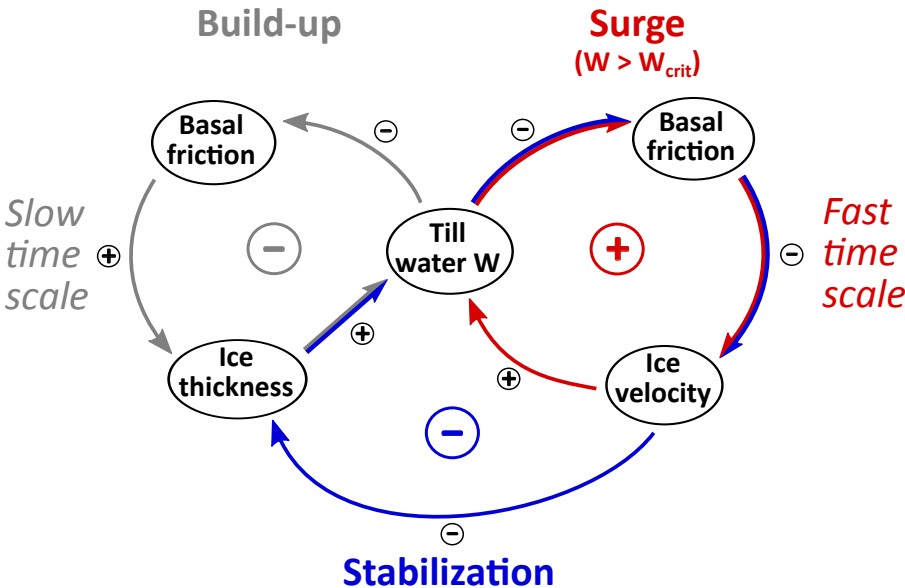

**Figure 2.** Schematic visualizing the three main feedback mechanisms, each of them dominating one of the three sub-sequent phases of slow ice build up (gray), abrupt surging (red) and stabilization (blue), forming a full surge cycle. The sign next to an arrow pointing from variable A to B indicates whether a small increase in variable A leads to an increase (+) or decrease (-) in variable B. According to this convention one can deduce from counting the negative links of a full loop whether this loop describes an amplifying (positive) or stabilizing (negative) feedback. An even number of negative links indicates a positive feedback loop (large +) whereas an odd number of negative links indicates a negative feedback loop (large -).

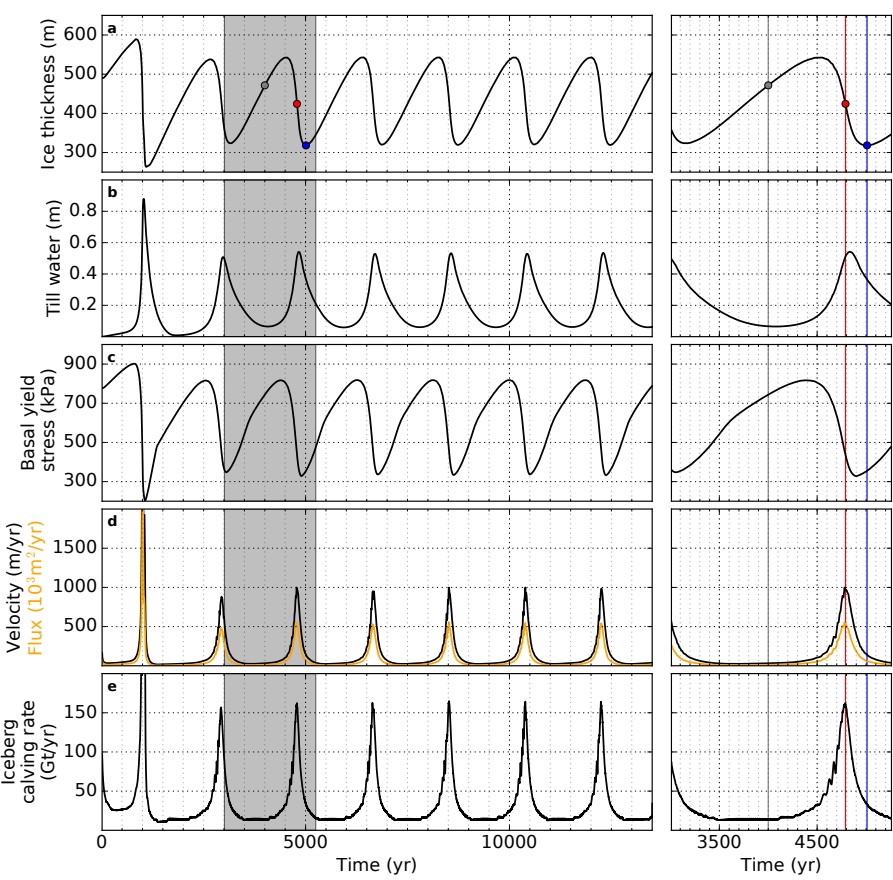

**Figure 3.** Timeseries of the main variables which characterize the feedback loops of growth, surge and stabilization in Fig. 2. **(a)** Ice thickness $H$, **(b)** till water thickness $W_{til}$, **(c)** basal yield stress $\tau_c$, **(d)** velocity and ice flux (orange), and **(e)** iceberg calving rate. Except for the calving rate, data shown is averaged over the area of grounded ice. The calving rate has been smoothed with a 200-year moving window. The right-hand-side of each panel shows a zoom into a full cycle (highlighted in gray). Colored circles in panel (a) show the points in time chosen to be representative for the phases of build-up (gray), surge (red) and stabilization (blue).

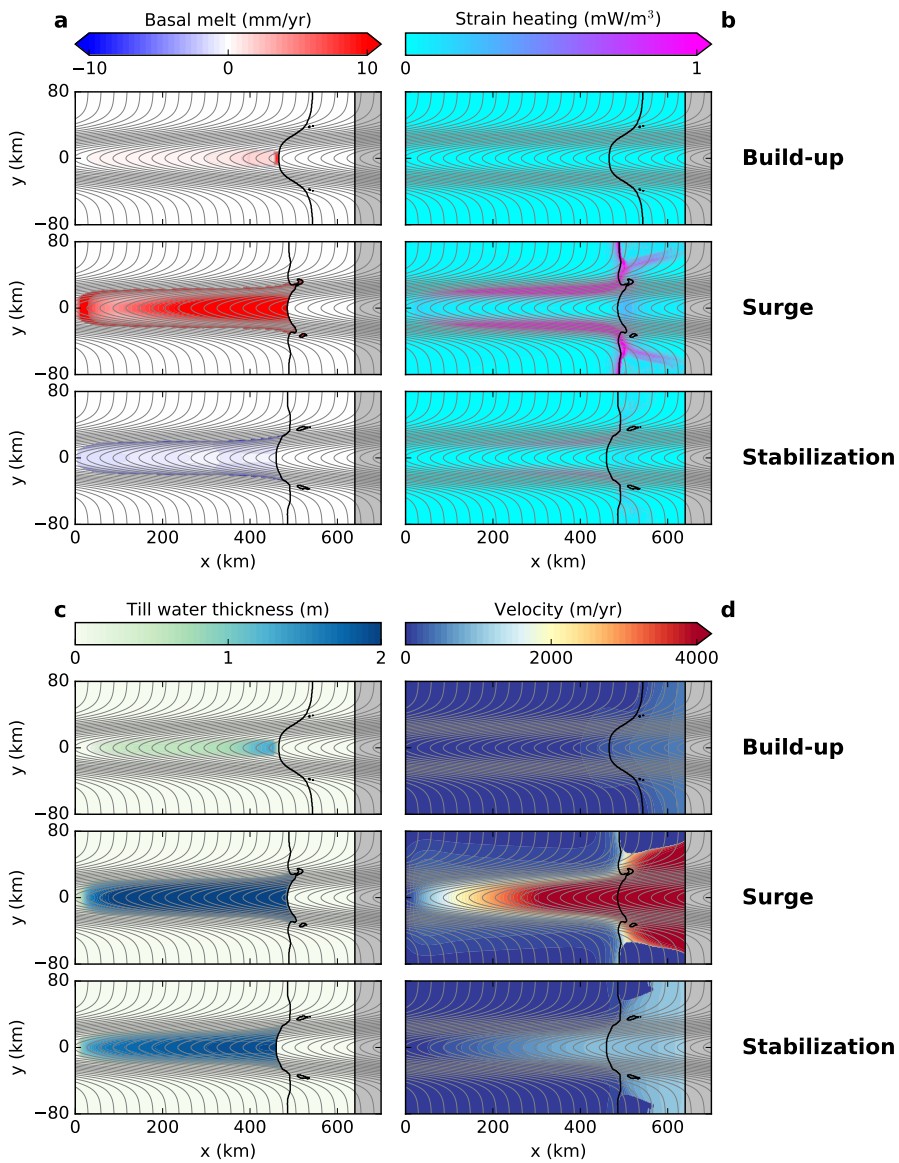

**Figure 4.** Fields of **(a)** basal melt rate $m$, **(b)** strain heating, **(c)** till water thickness $W_{til}$, and **(d)** velocity for a representative snapshot for each of the three phases of build-up, surge and stabilization (as denoted by the colored circles in Fig. 3). Thick black contours mark the grounding line and calving front. Bed topography shown by thin gray contours.

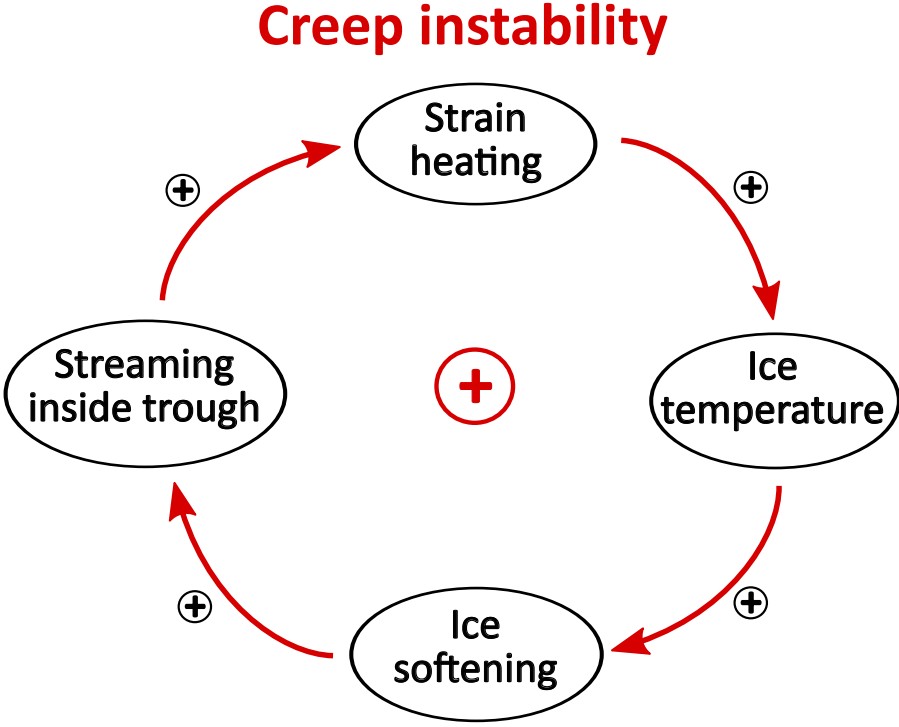

**Figure 5.** Positive feedback of creep instability, which fosters rapid ice streaming through the bed trough in addition to the positive feedback of ice-flow acceleration visualized in Fig. 2.

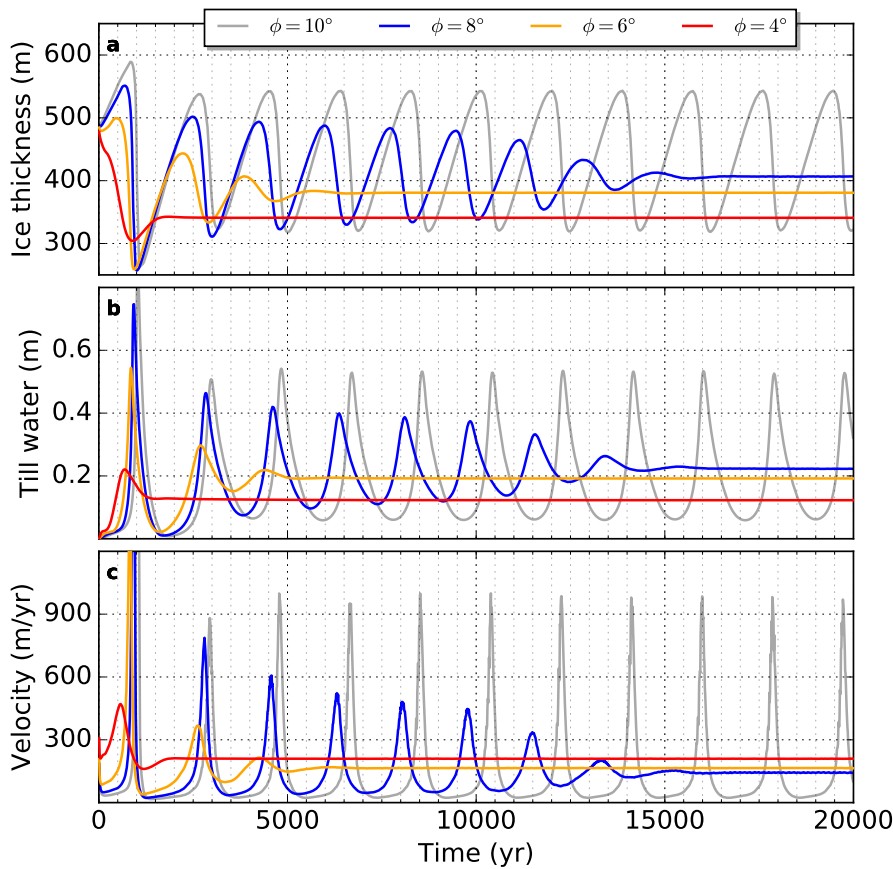

**Figure 6.** Timeseries of **(a)** ice thickness $H$, **(b)** till water thickness $W_{til}$, and **(c)** ice velocity (all averaged over area of grounded ice) for different values of the till friction angle ($\phi \leq 10$ °). Between $\phi = 10$ (default case) and $\phi = 8$ there is a transition from maintained cyclic surging to damped surging.

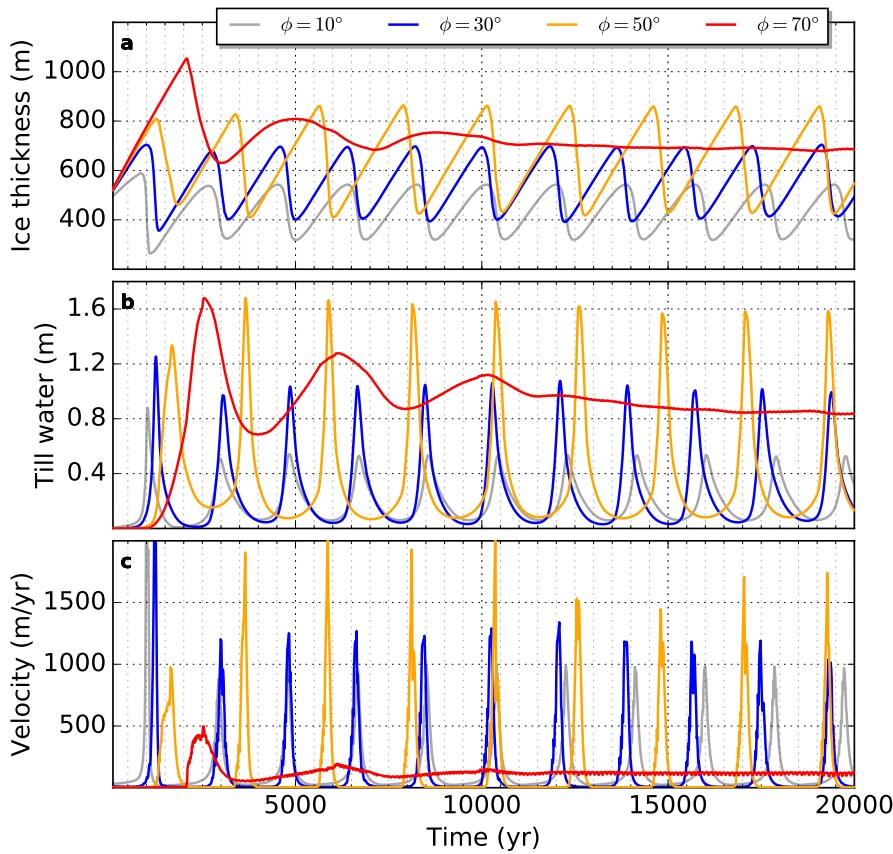

**Figure 7.** Timeseries analogous to Fig. 6, here for $\phi \geq 10$ °. For relatively large values of $\phi$ there is a transition from maintained cyclic surging to damped surging.

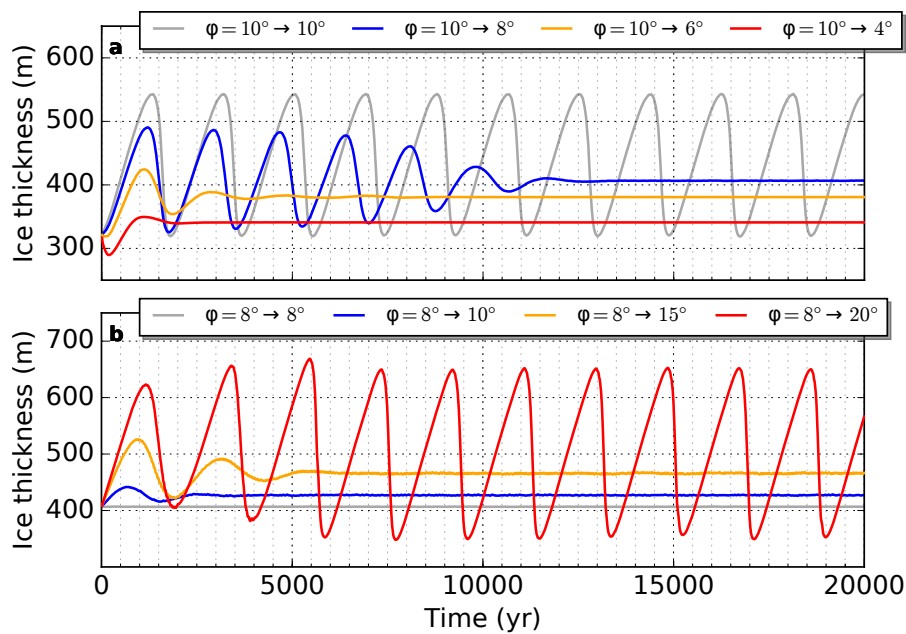

**Figure 8.** Timeseries of ice thickness $H$ for **(a)** an ice sheet in oscillatory equilibrium ($\phi = 10\,^\circ$) which is perturbed by a decrease of $\phi$ and **(b)** an ice sheet in stable equilibrium ($\phi = 8\,^\circ$) which is perturbed by an increase of $\phi$. In order to bring the stable ice sheet into the regime of maintained surging $\phi$ has to be increased substantially ($\phi = 8\,^\circ \to 20\,^\circ$), whereas it has to be lowered only slightly ($\phi = 10\,^\circ \to 8\,^\circ$) to stabilize the surging ice sheet.

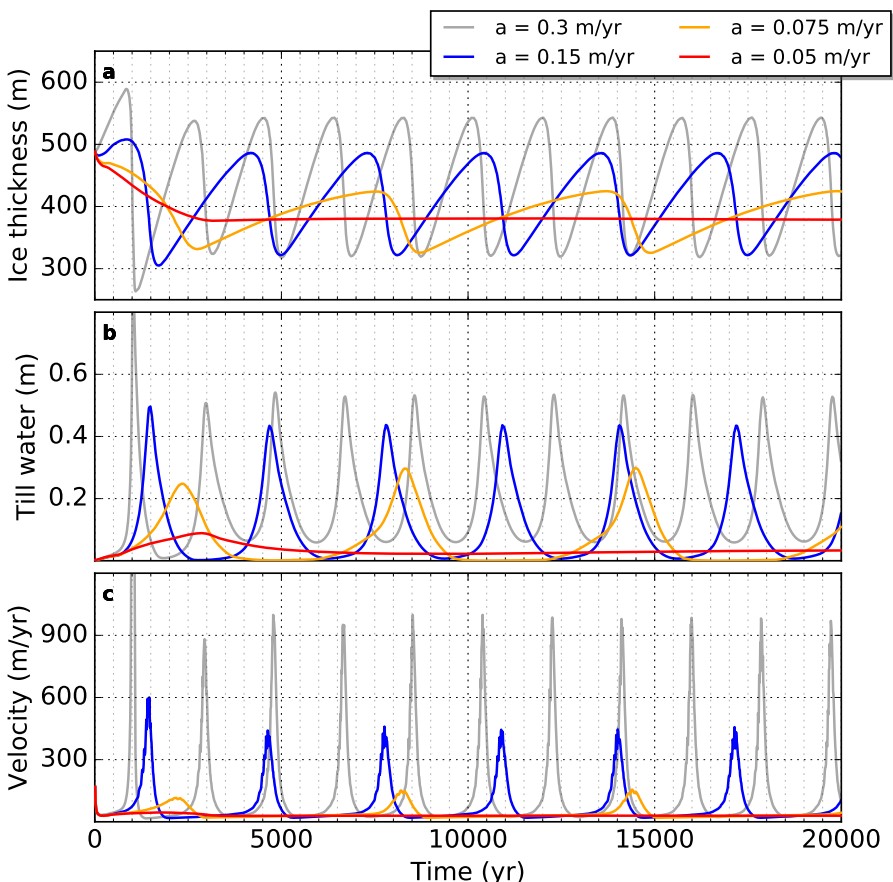

**Figure 9.** Timeseries of **(a)** ice thickness $H$, **(b)** till water thickness $W_{til}$, and **(c)** ice velocity (all averaged over area of grounded ice) for different values of the surface accumulation $a$. With decreasing $a$ (default case in gray) the surge magnitude decreases and the cycle duration increases such that for sufficiently low accumulation surging is not existent.

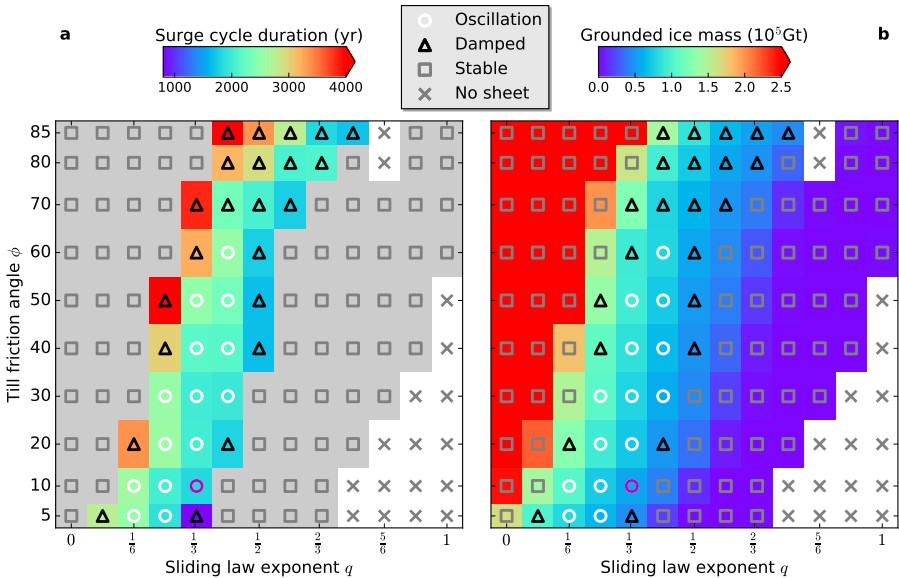

**Figure 10. (a)** Surge-cycle duration and **(b)** mean grounded ice mass for the $q - \phi$ parameter space. Each colored rectangle represents a simulation characterized by either oscillatory surging (white circles), damped surging (triangles) or stable equilibrium (squares). White rectangles with an "x" denote parameter combinations for which no grounded ice forms inside the bed trough and are thus not considered in the analysis. Since the simulations of stable ice flow do not exhibit periodicity by definition the associated rectangles in panel **(a)** are colored in gray. The default simulation with parameters of $q = \frac{1}{3}$ and $\phi = 10\,^\circ$ is highlighted by a purple circle (see Figs. 3 and A2, gray curve in Figs. 4, 6, 7, 8a and 9).

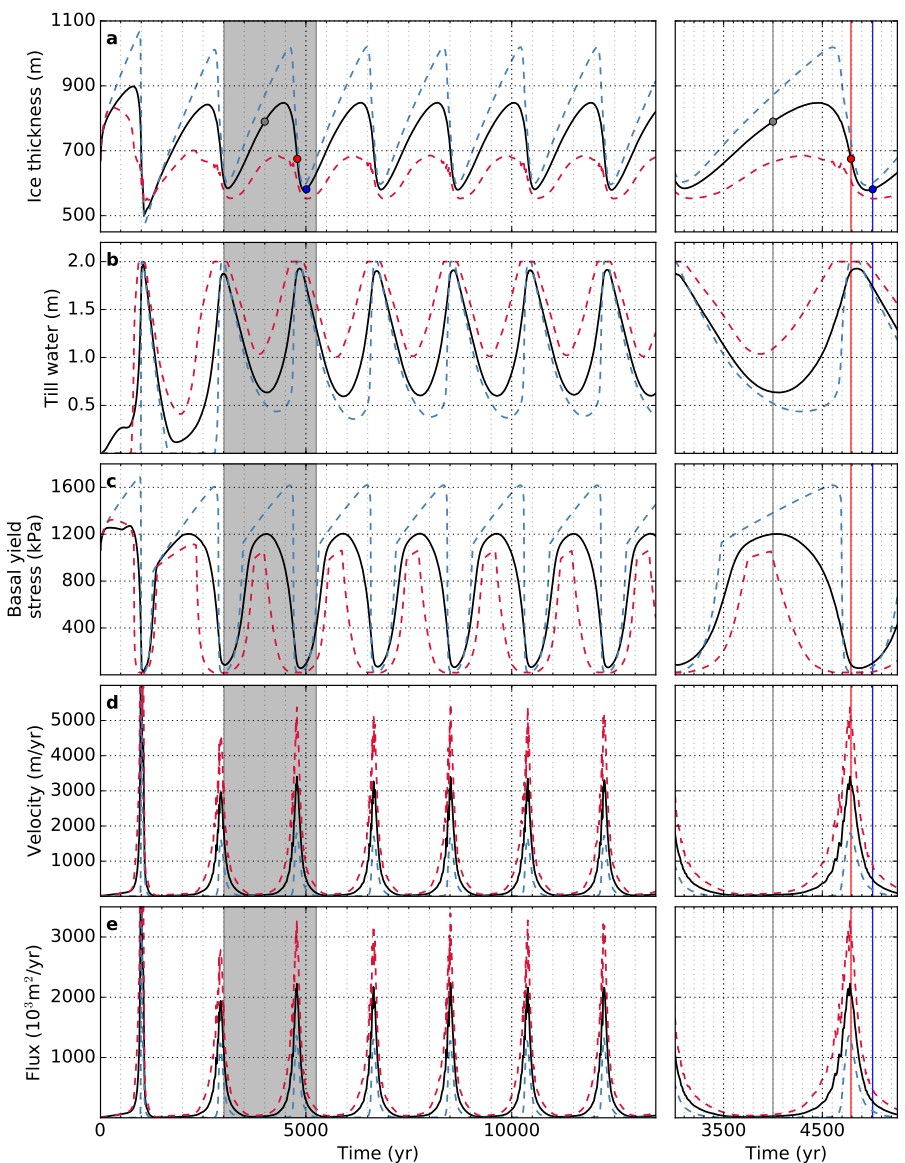

**Figure A1.** Timeseries complementing Fig. 3, showing centerline-averaged values (black curve) and two point measurements on the centerline (dashed blue and red curves; locations highlighted by squares in Fig. 1a) instead of grounded-area averaged values.

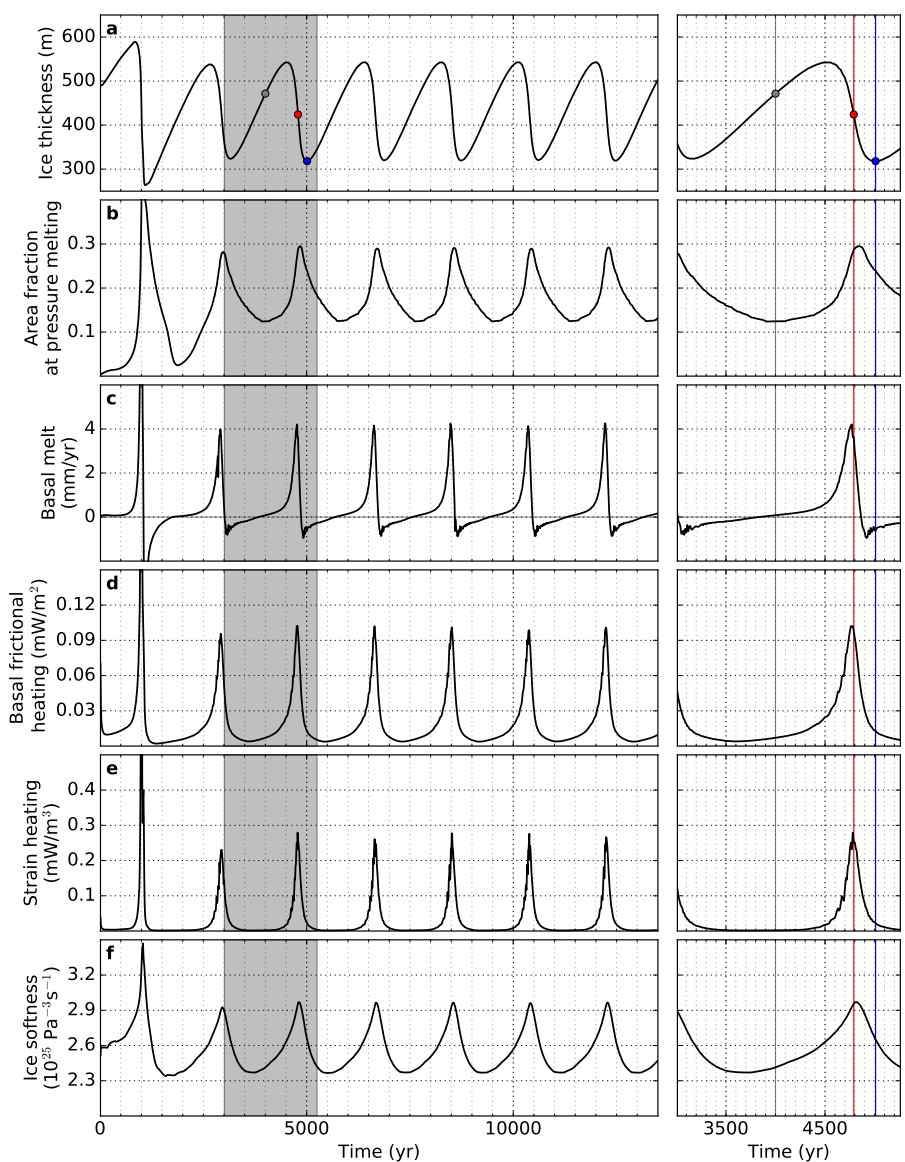

**Figure A2.** Additional timeseries to results in Fig. 3 of **(a)** ice thickness $H$, **(b)** fraction of grounded ice which is at pressure melting point at its base, **(c)** basal melt rate $m$, **(d)** basal frictional heating, **(e)** strain heating, and **(f)** vertically averaged ice softness. Except for panel (b) data shown is averaged over the area of grounded ice. The right-hand-sides of the panels are analogue to the ones in Fig. 3.

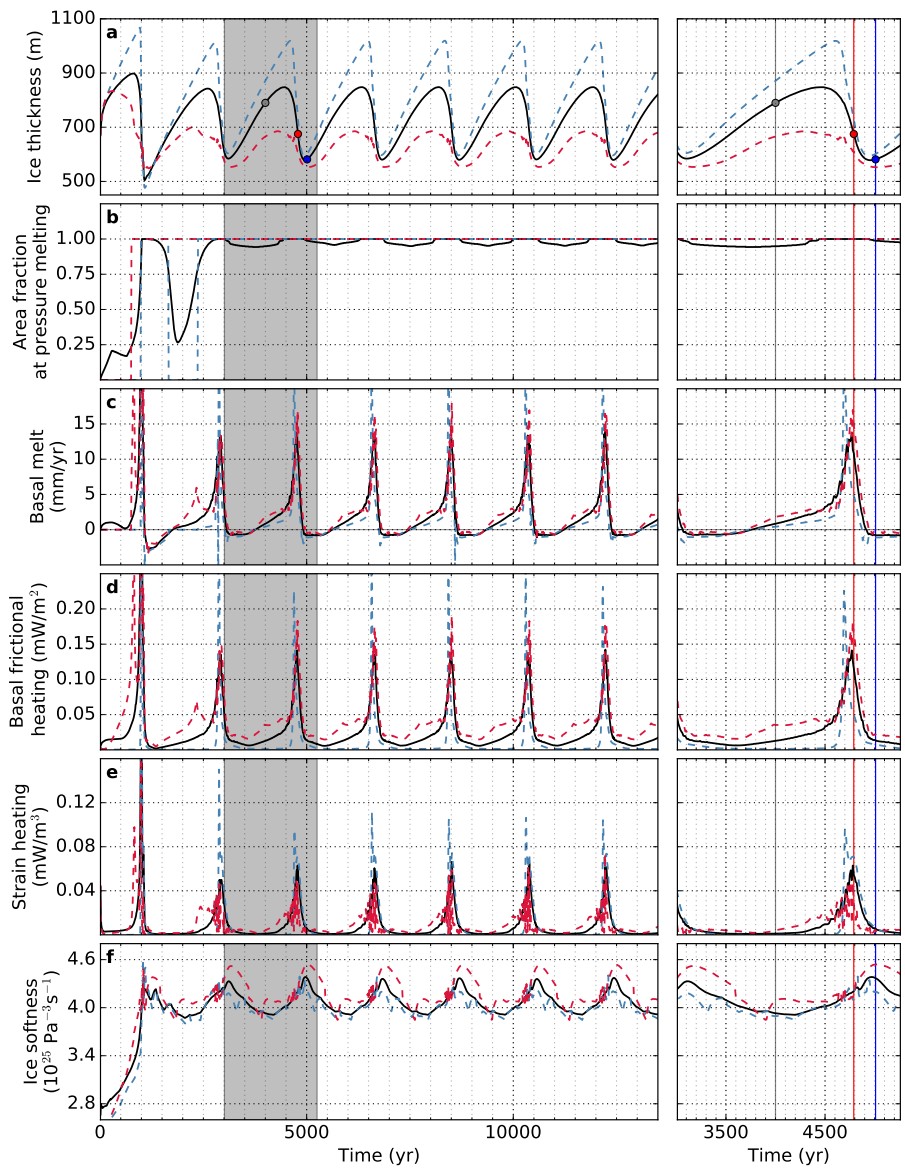

**Figure A3.** Timeseries complementing Fig. A2, showing centerline-averaged values (black curve) and two point measurements on the centerline (dashed blue and red curves; locations highlighted by squares in Fig. 1a) instead of grounded-area averaged values.