# Peer review of "From Heinrich Events to cyclic ice streaming: the grow-and-surge instability in the Parallel Ice Sheet Model"

_The Cryosphere, 2016_

## Referee Comment (RC1) · Anonymous Referee #1 · 21 Nov 2016

I provide a very cursory review because on initial quick skimming, it is clear that the manuscript is well constructed and that the scientific methodology supports well the conclusions drawn. The paper documents further an important form of ice flow variability that may have a bearing on how ice-flow developments in Antarctica are viewed in the future. (It would be interesting, for example, to address what is "really" the situation with Thwaites Glacier–a topic that is often linked with immediate effects of climate change. Attributing changes at Thwaites to just the developments of the last decade or 2 would bring one to question whether there were alternative explanations, e.g., is the outlet glacier subject to oscillations of the type shown in this manuscript.)

---

## Short Comment (SC1) · 9 Dec 2016

This is a well written paper on ice surging, and it is not the first one in this field. I enjoyed reading the paper as all findings are nicely presented. In the discussion of the results I missed a comparison of the results and mechanisms to those of other studies investigating parameter dependence of surge cycles in Heinrich events and related setups.

It would be interesting to know how the results with the more sophisticated sliding scheme differ from or support those obtained with the Shallow Ice Equation (e.g. Calov et al. (2002) and Greve et al. (2006)). Greve et al. (2006) study the dependence of the surge cycles on surface mass balance and basal friction coefficient. Are the mech-

anisms and time scale effects comparable (similar questions for Calov et al. (2010), where more models are taken into the comparison)?

Another paper that immediately comes to mind is the study by van Pelt and Oerlemans (2012), where the parameter dependence of surge cycles of a land-terminating glacier in a previous version of the same ice sheet model was studied. This calls for a comparison of the findings.

In the introduction, a mentioning of the full-stokes study of cyclic ice stream behavior by Kleiner and Humbert (2014) might be appropriate.

I'm looking forward to reading the final version of the paper. Please feel free to notify me when it is published. :)

Florian

Calov, R., Ganopolski, A., Petoukhov, V., Claussen, M., & Greve, R. (2002). Large-scale instabilities of the Laurentide ice sheet simulated in a fully coupled climate-system model. Geophys. Res. Lett., 29, 69–1–69–4. http://doi.org/10.1029/2002GL016078

Calov, R., Greve, R., Abe-Ouchi, A., Bueler, E., Huybrechts, P., Johnson, J. V., . . . (2010). Results from the Ice-Sheet Model Intercomparison Project–Heinrich Event INtercOmparison (ISMIP HEINO). Journal of Glaciology, 56(197), 371–383. http://doi.org/10.3189/002214310792447789

Greve, R., Takahama, R., & Calov, R. (2006). Simulation of large-scale ice-sheet surges : The ISMIP HEINO experiments. Polar Meteorology and Glaciology, 20, 1–15. Retrieved from http://hdl.handle.net/2115/29659

Kleiner, T., & Humbert, A. (2014). Numerical simulations of major ice streams in western Dronning Maud Land, Antarctica, under wet and dry basal conditions. Journal of

Glaciology, 60(220), 215–232. http://doi.org/10.3189/2014JoG13J006

Van Pelt, W. J. J., & Oerlemans, J. (2012). Numerical simulations of cyclic behaviour in the Parallel Ice Sheet Model (PISM). Journal of Glaciology, 58(208), 347–360. article. http://doi.org/10.3189/2012JoG11J217

---

## Referee Comment (RC2) · S. L. Cornford (Referee) · 23 Dec 2016

This paper describes and application of the very respectable PISM ice sheet model to idealized simulations of surge cycles in marine ice streams. It differs from other studies in the same sort of area by modelling both longitudinal and lateral stresses (as opposed to just one or neither) without parametrizations. I think it has the basis of a good paper, but I think it needs an extra result or two to make it a really good paper. Ideally, I would have like to have seen the experimental design include the MISMIP+ reverse slopes (where buttressing or the lack of really matters), but I think that might be too much to ask for. So instead, I'd like to suggest that the authors also carry out some simulations with the linear sliding law. On page 6, I see that the paper suggests that either the

nonlinear sliding law or the more natural treatment of buttressing is responsible for differences with the Robel 2016 paper (linear sliding, parametrized buttressing). By choosing a basal traction coefficient such that the ice sheet is comparable (e.g. same GL position at furthest advance), this can be tested in some more detail (I.e, if the results are still different, it must be the buttressing treatment)

General Comments ————————-

The feedback diagrams are a nice idea to make the subject easier to understand. I wonder if the first of these figures (and the text that describes it) needs a little work. It is not so difficult to understand that there are some negative feedbacks e.g H → + → V → - → H and some positive feedbacks e.g W → + → V → + → W . but the key to all of this is in the detail of when and why one dominates. I don't really read that from the diagrams. Also, there is a mix of degree-of-freedom variables (H,V,W) and derived quantities (basal traction, flux), I think this could be simplified.

I wanted to read some discussion of the relationship between the various equations and time scales comes about (e.g, what is the source of the 1.8 ky scale – the drainage rate, or the time taken to advect cold ice from the divide, or something else. Should it be a surprise that it is not much affected by SSA stresses, which tend to have limited importance far upstream from the GL)

I'm not sure about the stabilization phase (P5, L17) being a separate negative feedback system (blue loop). First,it has the same time scale as the surge phase. My naïve reading of this is that at some point, the thinner colder ice means that melt-rate starts to drop, so that dW/dt < 0, then the same positive feedback that caused the surge )ie W → + → V → + → W works in reverse (W →- → V → - -> W). I'm no surge expert though – do other authors agree with you?

The surge-damping results are interesting, I think you could extend perhaps them . At the moment you have undamped surging (phi = 10) and decay to states that maintain a steady thin ice stream (phi <= 8), where presumably the bed is not frozen. Do steady

'thick and slow' systems occur when phi » 10. Like wise, it would be interesting to see what happened if you switch to phi = 10 from the phi = 8 system.

The manuscript seems to somewhat over-rate its novelty e.g

(1) abstract, 'we identify .. the central feedbacks' – that's a big claim. Surely others have noted the same.

(2) P2, L10 "In particular, and in contrast to many of the previous studies, our simulations use a sliding law that is based on the stress balance of the ice and thereby has stress boundary conditions."

Some papers have considered non-linear sliding, membrane stresses, etc in studies of thermo-mechanical instabilities. Obvious examples include Hindmarsh, G.RL, 2009 which is not cited, and Beuler and Brown 2009 (which is cited), which also describes the original version of the SSA/SIA scheme and much else regarding the PISM model used here, the major exception being Aschwandens 2012 improved PISM thermodynamics scheme. OK, the "many" makes P2,L10 technically true, but this is not the only statement of this sort, the cumulative effect is to appear to be claiming too much.

(3) The connection to Heinrich events, with a ice plus basal water model (not such a nice one) is described at length in Roberts et al, Clim. Past, 12, 1601 (doi:10.5194/cp-12-1601-2016)

Specific comments ──────────────

P2, L31 "A linear interpolation of the freely evolving grounding line and accordingly interpolated basal friction enable realistic grounding-line motion similar to models of higher order (Feldmann et al., 2014)."

I don't think Feldmann 2014 shows this, exactly. The interpolation may represent a modest improvement but the time-dependent behaviour in Feldmann 2014 is clearly not close to convergent unless the mesh is resolved to around 1-2 kilometers., and indeed, the *non-interpolated* (model A) results at around 1km have features seen in

demonstrably resolved SSA (see the MISMIP3d paper) and Stokes (see Gagliardini 2016) models that the interpolated (model B) results lack . Probably the SSA/SIA physics and 1 km resolution chosen in this paper is adequate, but Feldmann 2014 is not the main reason even if it helps. You could say

"A linear interpolation of the freely evolving grounding line and accordingly interpolated basal friction, together with the use of one-sided differences* in the driving stress close to the GL, permit SSA physics to be treated with mesh resolutions of around 1 km (Feldman et al 2014)".

*Correct? I thought you did this. I do too because I found it made a big difference, e.g (sorry to mention my own papers) [Cornford 2013 http://dx.doi.org/10.1016/j.jcp.2012.08.037] .whereas the interpolation helped only a bit [Cornford 2016] https://doi.org/10.1017/aog.2016.13.

P4, L7 "The superposition of both components yields a bed trough which is symmetric in both x and y directions" → reflection symmetric about y = 0, but no x-symmetry in the formulas given. I think (from other parts of the paper, that you meant symmetry about x = 0 so instead of b(x) you have b(|x|)? however, you could just say that a reflection condition (dh/dx = 0, u = 0, dv/dx = 0) is satisfied at x = 0

P4, L10 "Resulting convergent flow and associated horizontal shearing enable the emergence of ice-shelf buttressing, having a stabilizing effect on the grounding line...". Not really "stabilizing" – even with no ice shelf there are no obvious unstable equilibria of the MISI sort in this geometry. Presumably the steady GL is further downstream than it might if the shelf was removed.

P5, L33, "...explained by assuming that a thinner ice sheet before the surge leads to a less dramatic surge [fine by me] and thus to a larger minimum [not fine by me]". A less dramatic surge starting from a thinner sheet could lead to the same finial thickness as a more dramatic surge starting from a thicker sheer, or pretty much any other combination.

Fig 7. The frequency (w) and amplitude (A) of surges decays with a. Seems like there might be a critical a between 0.05 and 0.075 where the surging is turned on/off. I wonder how w, A behave around that point? That might be an unreasonable request, depending how long the model takes to run.

---

## Author Comment (AC1) · 24 Apr 2017

The comment was uploaded in the form of a supplement:
http://www.the-cryosphere-discuss.net/tc-2016-235/tc-2016-235-AC1-supplement.zip
* * *

---

## Author Response (AR1)

**Detailed response to the editor on manuscript tc-2016-235**

"From Heinrich Events to cyclic ice streaming: the grow-and-surge instability in the Parallel Ice Sheet Model"

by J. Feldmann and A. Levermann

Dear Prof Vieli,

We would like to thank you for handling the review process and the reviewers for their detailed look at our manuscript. We are happy for the very positive assessment of the Reviewer #1 and the recommendation of the publication of our manuscript. We would also like to thank Reviewer #2 for the overall positive review and the very constructive and helpful comments. Following the reviewer's main request, i.e., to expand our results section, we carried out numerous additional simulations and we think that the new results coming out of them definitely add to our study. Also we are confident that with the revision of our manuscript we address the other issues raised by Reviewer #2. Three new figures (Figs. 7, 8 and 10) and a new section (Sec. 3.3, "Role of basal sliding law") have been added to the manuscript. Revising the manuscript, we also took into account the valuable recommendations given in a short comment by a third reviewer. Last but not least, we picked up the suggestions by the Editor to discuss our prescribed calving condition (p. 4, l. 15-19) and also elaborate on the rather large yield stresses found in our experiments (p. 6, l. 1-12).

We would like to highlight that our new simulations include, but are not limited to, a parameter study in which we explore 1) the role of the basal sliding law and 2) the influence of bed strength on the surge dynamics. Though this was not requested by Reviewer #2 it covers several of his requests. In our simulations the sliding law was fixed (sliding law exponent $q$=1/3) but now spans the range from purely plastic sliding ($q$=0) to linear sliding ($q$=1). Also the bed roughness was rather confined to a small set of parameter values but now a wide range from very slippery to rough bed conditions is represented in our study. The resulting two-dimensional parameter space in particular allows to infer the conditions that promote or inhibit surging as well as a discussion of the time scale of the surge cycle in our simulations.

Please find below the *reviewers' comments in italics* and our detailed response in blue. We have further attached a revised manuscript that highlights the changes in the submission, as well as a clean revised version.

Best wishes,
J. Feldmann and A. Levermann

**Interactive comment on "From Heinrich Events to cyclic ice streaming: the grow-and-surge instability in the Parallel Ice Sheet Model"**
**by**
**Johannes Feldmann and Anders Levermann**

Anonymous Referee #1

*I provide a very cursory review because on initial quick skimming, it is clear that the manuscript is well constructed and that the scientific methodology supports well the conclusions drawn. The paper documents further an important form of ice flow vari-ability that may have a bearing on how ice-flow developments in Antarctica are viewed in the future. (It would be interesting, for example, to address what is "really" the situa-tion with Thwaites Glacier–a topic that is often linked with immediate effects of climate change. Attributing changes at Thwaites to just the developments of the last decade or 2 would bring one to question whether there were alternative explanations, e.g., is the outlet glacier subject to oscillations of the type shown in this manuscript.)*

We would like to thank the reviewer for the effort to review our manuscript and are glad for the very positive assessment. We are pleased that the reviewer recommends our paper for publication. The question whether glaciers like Thwaites Glacier show large-scale instability (marine ice sheet instability) or are rather subject to oscillations comparable to these in our simulations would indeed by a very interesting one. It might be an exciting topic for future work.

**Interactive comment on "From Heinrich Events to cyclic ice streaming: the grow-and-surge instability in the Parallel Ice Sheet Model"**
by
Johannes Feldmann and Anders Levermann

S. L. Cornford (Referee)
s.l.cornford@bristol.ac.uk

*This paper describes and application of the very respectable PISM ice sheet model to idealized simulations of surge cycles in marine ice streams. It differs from other studies in the same sort of area by modelling both longitudinal and lateral stresses (as opposed to just one or neither) without parametrizations. I think it has the basis of a good paper, but I think it needs an extra result or two to make it a really good paper. Ideally, I would have like to have seen the experimental design include the MISMIP+ reverse slopes (where buttressing or the lack of really matters), but I think that might be too much to ask for. So instead, I'd like to suggest that the authors also carry out some simulations with the linear sliding law.*

We would like to thank the reviewer for the positive assessment of our manuscript and the very constructive and helpful comments. As asked for by the reviewer, we carried out a several sets of additional simulations and are confident that the results coming out of them further enrich our study. The reviewer's main request to also run experiments that use the linear version of the applied sliding law inspired us to carry out a parameter study which explores the full range of the sliding exponent *m* in the sliding law (Eq. 1). Previously this parameter was fixed in our simulations (*q*=1/3) but now spans the range from *q*=0 (purely plastic sliding law) to *q*=1 (linear sliding law). Another suggestion by the reviewer (see next reviewer comment) was to carry out further simulations with a modified basal traction coefficient, i.e., the till friction angle *phi*. Consequently we vary *phi* covering the range from very slippery to very rough bed conditions for each of the prescribed values of *m* which allows us to explore the *q-phi* parameter space. The analysis of the results is done in the newly added Section 3.3, including the location of the different flow regimes within the *q-phi* space and an investigation of the influence of both parameters on the period duration of the surge cycle and ice volume (see new Fig. 10). In the revised discussion/conclusion section we discuss the new findings also in the light of results from other studies (p. 9, l. 3-31).

*On page 6, I see that the paper suggests that eitherthe nonlinear sliding law or the more natural treatment of buttressing isresponsible for differences with the Robel 2016 paper (linear sliding, parametrized buttressing). By choosing a basal traction coefficient such that the ice sheet is comparable (e.g. same GL position at furthest advance), this can be tested in some more detail (I.e, if the results are still different, it must be the buttressing treatment)*

One outcome of our parameter-space investigation is that in the particular case of linear sliding (*q*=1) the ice sheet takes on a very different shape compared to the *q*=1/3 case in our simulations. There is few to no grounded ice inside the channel (mostly ice shelf) and only a few 10 m of grounded ice thickness outside the channel which. This qualitatively different behavior is independent of the friction parameter *phi* and thus the comparison suggested by the reviewer is not feasible for the specific setup.
Motivated by the fact that the ice sheet is generally very thin for large *q* (particularly in the linear case), we conducted further simulations with increased surface accumulation. However, even in simulations which use an accumulation rate which is up to 10 times larger than the default value,

barely any grounded ice forms inside the channel. The evolution of a proper ice sheet comparable to the ice sheet in the mentioned study of Robel et al. 2016 for the particular case of $m=1$ might be achieved by the exploration of further model parameters (e.g. drainage rate $C\_d$ or basal velocity scaling parameter $u\_0$). However, given the amount of simulations we already conducted for this study, this would be clearly beyond our means and we hope for the understanding of the reviewer.

*General Comments —————————-*

*The feedback diagrams are a nice idea to make the subject easier to understand. I wonder if the first of these figures (and the text that describes it) needs a little work. It is not so difficult to understand that there are some negative feedbacks e.g*
*$H \to + \to V \to - \to H$*
*and some positive feedbacks e.g*
*$W \to + \to V + \to W$*
*but the key to all of this is in the detail of when and why one dominates. I don't really read that from the diagrams. Also, there is a mix of degree-of-freedom variables (H,V,W) and derived quantities (basal traction, flux), I think this could be simplified.*

We are glad that the reviewer likes our idea of using feedback diagrams to visualize the main feedback mechanism. We fully agree with the reviewer that our feedback loops include derived variables. For instance, the basal shear stress $tau\_b$ is a quantity derived from velocity $V$, i.e., $tau\_b=f(V)$ (see Eq. 1). However, at same time $V$ can also be understood as a function of $tau\_b$ since $tau\_b$ has a strong influence on $V$ through the SSA equation. Thus, when drawing the feedback loops in Fig. 2, we do not want to claim them to be of the mathematical exactness of, e.g., Feynman diagrams, but consider them as an illustration of the main mechanisms in the presented surge simulations, including the variables that we find to be most relevant. Leaving out or introducing additional (derived) variables would simplify the loop (at the expense information loss) or add complexity to it. A suitable analogy which came to our mind is the sea ice-albedo feedback, stating that more ice area (*A*) leads to higher albedo (*alpha*) which in turn leads to a larger ice area. The simple positive feedback loop then would read:
$A \to + \to alpha \to + \to A$.
However, if one is also interested in the role of ice temperature (*T*) one could add it to the loop:
$A \to + \to alpha \to - \to T \to - \to A$.
Though *T* can be regarded as a quantity derived from *alpha* and thus might be regarded as redundant for the overall feedback, its introduction adds detail to the loop, shedding light onto the physics that are behind the connection between *alpha* and *A*. We think that the same holds for the basal traction in the two upper loops of our Fig. 2. Leaving it out in our view would oversimplify the diagrams since the basal friction takes a very relevant role in connecting till water and ice velocity/thickness. We thus would like to keep the two upper loops as they are but only modify the bottom loop (see our comments below).

*I'm not sure about the stabilization phase (P5, L17) being a separate negative feedback system (blue loop). First,it has the same time scale as the surge phase. My naïve reading of this is that at some point, the thinner colder ice means that melt-rate starts to drop, so that dW/dt < 0, then the same positive feedback that caused the surge )ie*
*$W \to + \to V \to + \to W$ works in reverse ($W \to - \to V \to - \to W$). I'm no surge expert though – do other authors agree with you?*

After an in-depth discussion of this issue we came to the conclusion that the surge loop indeed

also plays a role during the stabilization phase, as soon as the till water has reached its maximum and starts to drop (red loop in reverse, as suggested by the reviewer). However, we are convinced that the stoppage of self-enforced surging requires a counteracting negative feedback which has to be in effect simultaneously with the positive surging feedback which leads to stabilization. This would not be the case when only considering the self-enforcing (red) loop since the change of sign in *dW/dt* mentioned by the reviewer could not be realized (during surging till water would simply grow and grow since the feedback is self-enforcing).

We think that during the surge phase the effect of the velocity increase on the ice-sheet thickness forms the negative feedback that is required to counteract the surge feedback and hence is responsible for the stabilization (blue feedback loop). This feedback loop is indeed in accordance with the reviewer's reading of the processes: increasing ice velocity leads to smaller overall ice thickness which means less till water production (via lower basal melt rate). Through larger basal friction the ice flow acceleration decreases and the ice thickness can stabilize. The difference in the time scale between the red and blue loops lies in the faster response of the till water to a velocity increase compared to the relatively long time it takes until the velocity-driven discharge has thinned the ice sheet sufficiently (and which then cools, as mentioned by the reviewer) such that the melt rate drops (and thus till water) and the ice sheet can stabilize. As requested by the reviewer, we now go into this in more detail in the text (p. 5, l. 25-27 and 29-31). To simplify Fig. 2 we removed the derived quantity ice flux *Q* (see reviewer comment above) in the bottom of the figure. As mentioned above, we agree with the reviewer that also the surge loop (in reverse fashion) is at play during the stabilization which we now make clear in the text and would offer to additionally put a blue arrow between *V* and *W*. However, preferably and for the sake of simplicity we would like to leave the figure in the revised submitted form.

*I wanted to read some discussion of the relationship between the various equations and time scales comes about (e.g, what is the source of the 1.8 ky scale – the drainage rate, or the time taken to advect cold ice from the divide, or something else. Should it be a surprise that it is not much affected by SSA stresses, which tend to have limited importance far upstream from the GL)*

It is indeed worthwhile to have a more detailed discussion of the time scale as pointed out by the reviewer. Our revised conclusion/discussion section now discusses the surge time scale dependent on the examined bed strength, surface accumulation and sliding-law exponent, and includes a comparison to time scales found in other studies. In the new Sec. 3.2 now we also give a physical reasoning on how the above mentioned variables affect the time scale (p. 7, l. 27-33), discussing their role in the sliding law, the basal model (Eqs. 1 and 2) and the shallow-shelf approximation of the stress balance.

*The surge-damping results are interesting, I think you could extend perhaps them . At the moment you have undamped surging (phi = 10) and decay to states that maintain a steady thin ice stream (phi <= 8), where presumably the bed is not frozen. Do steady 'thick and slow' systems occur when phi » 10.*

Looking also at the other end of the parameter range of *phi*, as suggested by the reviewer, makes a lot of sense. The investigation of large values of *phi* is covered by our added parameter study which reveals that there exists indeed a regime of stable flow of a rather thick ice sheet. To visualize the surge damping for *phi » 10* we included a timeseries analogous to the one for the *phi <= 8* regime (see new Fig. 7) which is briefly analyzed in the results section (p. 6, l. 21-26).

*Like wise, it would be interesting to see what happened if you switch to phi = 10 from the phi = 8 system.*

This is indeed a very nice idea! We carried out such switching experiment for both directions (perturbing from oscillatory state into stable equilibrium and vice versa). The outcome is that the ice sheet in stable equilibrium requires a comparatively large perturbation (*phi* = 8 → 20 and not 8 → 10 as one could expect from the spinup experiments) in order to turn into a state of maintained surging. In contrast, a small perturbation is sufficient to bring the continuously oscillating ice sheet into a stable steady state, i.e., *phi* = 10 → 8. The results are visualized in the new Fig. 8 and analyzed analyzed in p. 6, l. 27 – p. 7, l. 4.

*The manuscript seems to somewhat over-rate its novelty e.g*
*(1) abstract, 'we identify .. the central feedbacks' – that's a big claim. Surely others have noted the same.*

We agree with the reviewer that the term identify could be misunderstood and thus removed it.

*(2) P2, L10 "In particular, and in contrast to many of the previous studies, our simulations use a sliding law that is based on the stress balance of the ice and thereby has stress boundary conditions."*
*Some papers have considered non-linear sliding, membrane stresses, etc in studies of thermo-mechanical instabilities. Obvious examples include Hindmarsh, G.RL, 2009 which is not cited, and Beuler and Brown 2009 (which is cited), which also describes the original version of the SSA/SIA scheme and much else regarding the PISM model used here, the major exception being Aschwandens 2012 improved PISM thermodynamics scheme. OK, the "many" makes P2,L10 technically true, but this is not the only statement of this sort, the cumulative effect is to appear to be claiming too much.*

We thank the reviewer for his advice and understand his concern. We have substituted the word "many" by "several" here (p. 2, l. 10 and 15) and also in the Methods section (p. 3 l. 6) in order to not appear overstating but at the same time account for the fact that there are a bunch of studies out there that use a much simpler representation of basal sliding. We also thank the reviewer for the additional reference, which we unintentionally did not include when writing the manuscript. We now cite Hindmarsh 2009 in the Methods section (p. 3 l. 5/6).

*(3) The connection to Heinrich events, with a ice plus basal water model (not such a nice one) is described at length in Roberts et al, Clim. Past, 12, 1601 (doi:10.5194/cp-12-1601-2016)*

We thank the reviewer for this reference, which we now cite in the introduction and in the discussion section. There we also clarify that in contrast to other studies our simulations do not capture the characteristic time scale at which Heinrich Events take place (p. 9, l. 3-8).

*Specific comments ————————*

*P2, L31 "A linear interpolation of the freely evolving grounding line and accordingly interpolated basal friction enable realistic grounding-line motion similar to models of higher order (Feldmann et al., 2014)."*
*I don't think Feldmann 2014 shows this, exactly. The interpolation may represent a modest improvement but the time-dependent behaviour in Feldmann 2014 is clearly not close to*

*convergent unless the mesh is resolved to around 1-2 kilometers., and indeed, the \*non-interpolated\* (model A) results at around 1km have features seen in demonstrably resolved SSA (see the MISMIP3d paper) and Stokes (see Gagliardini 2016) models that the interpolated (model B) results lack . Probably the SSA/SIA physics and 1 km resolution chosen in this paper is adequate, but Feldmann 2014 is not the main reason even if it helps. You could say*
*"A linear interpolation of the freely evolving grounding line and accordingly interpolated basal friction, together with the use of one-sided differences\* in the driving stress close to the GL, permit SSA physics to be treated with mesh resolutions of around 1 km (Feldman et al 2014)".*
*\*Correct? I thought you did this. I do too because I found it made a big difference, e.g (sorry to mention my own papers) [Cornford 2013 http://dx.doi.org/10.1016/j.jcp.2012.08.037] .whereas the interpolation helped only a bit [Cornford 2016] https://doi.org/10.1017/aog.2016.13.*

We thank the reviewer for the scrutiny in reading our manuscript. The use of one-sided differences in the driving stress is an important detail that we missed to mention. As suggested by the reviewer we modified the phrase accordingly and at the same formulate our statement in a less claiming manner (p.3 , l. 2-4).

*P4, L7 "The superposition of both components yields a bed trough which is symmetric in both x and y directions" → reflection symmetric about y = 0, but no x-symmetry in the formulas given. I think (from other parts of the paper, that you meant symmetry about x = 0 so instead of b(x) you have b(|x|)? however, you could just say that a reflection condition (dh/dx = 0, u = 0, dv/dx = 0) is satisfied at x = 0*

We thank the reviewer for pointing this inconsistency in the setup description. In the formula for the x component of the bed topography (p. 4, l. 7) b(x) should indeed read b(|x|), which we corrected. To be more precise now we also mention the symmetry axes in x and y direction, respectively (p. 4, l. 11).

*P4, L10 "Resulting convergent flow and associated horizontal shearing enable the emergence of ice-shelf buttressing, having a stabilizing effect on the grounding line...". Not really "stabilizing" – even with no ice shelf there are no obvious unstable equilibria of the MISI sort in this geometry. Presumably the steady GL is further downstream than it might if the shelf was removed.*

We agree with the reviewer that in our simulations buttressing does not have a stabilizing effect in the sense of inhibiting a MISI. We thus modified the phrase as suggested by the reviewer (p. 4, l. 13-14).

*P5, L33, "...explained by assuming that a thinner ice sheet before the surge leads to a less dramatic surge [fine by me] and thus to a larger minimum [not fine by me]". A less dramatic surge starting from a thinner sheet could lead to the same finial thickness as a more dramatic surge starting from a thicker sheer, or pretty much any other combination.*

This line of thought might indeed be a bit speculative and thus we removed the paragraph. We added two statements to the text regarding the cycle duration and till water thickness which might be easily drawn out of Fig. 6 but in our opinion are worth also to be mentioned in the text (p. 6, l. 17-18).

*Fig 7. The frequency (w) and amplitude (A) of surges decays with a. Seems like there might be a critical a between 0.05 and 0.075 where the surging is turned on/off. I wonder how w, A behave*

*around that point? That might be an unreasonable request, depending how long the model takes to run.*

This might be indeed another interesting thing to look into. However, further simulations are beyond our means and we hope for the understanding of the reviewer.

**Interactive comment on "From Heinrich Events to cyclic ice streaming: the grow-and-surge instability in the Parallel Ice Sheet Model"**
**by**
**Johannes Feldmann and Anders Levermann**

F. Ziemen
florian.ziemen@mpimet.mpg.de

*This is a well written paper on ice surging, and it is not the first one in this field. I enjoyed reading the paper as all findings are nicely presented.*

We would like to thank the reviewer for the positive assessment of our manuscript and the constructive comments which we address below.

*In the discussion of the results I missed a comparison of the results and mechanisms to those of other studies investigating parameter dependence of surge cycles in Heinrich events and related setups. It would be interesting to know how the results with the more sophisticated sliding scheme differ from or support those obtained with the Shallow Ice Equation (e.g. Calov et al. (2002) and Greve et al. (2006)). Greve et al. (2006) study the dependence of the surge cycles on surface mass balance and basal friction coefficient. Are the mechanisms and time scale effects comparable (similar questions for Calov et al. (2010), where more models are taken into the comparison)? Another paper that immediately comes to mind is the study byvan Pelt and Oerlemans (2012), where the parameter dependence of surge cycles of a land-terminating glacier in a previous version of the same ice sheet model was studied. This calls for a comparison of the findings.*

We are glad for this helpful hint and added a paragraph to the manuscript, discussing the time scale of the surging in our simulations and comparing it to other studies, including the ones suggested above (p. 8, l. 3-31).

*In the introduction, a mentioning of the full-stokes study of cyclic ice stream behavior by Kleiner and Humbert (2014) might be appropriate.*

Thanks for the hint. We now include the reference in the introduction (p. 2, l. 8-10).

*I'm looking forward to reading the final version of the paper. Please feel free to notify me when it is published. :)*

*Florian*

[revised manuscript text omitted]

---

## Referee Report (RR1)

In their manuscript *From Heinrich events to cyclic ice streaming, the grow-and-surge instability in the Parallel Ice Sheet model*, Johannes Feldmann and Anders Levermann nicely describe their findings from surge-type experiments with the Parallel Ice Sheet Model. In this respect, this manuscript is very similar to the Van Pelt and Oerlemans 2012 publication. There is nothing fundamentally wrong with this. Testing previously obtained results is an important process in science, and it is interesting to see that and under which parameters the current version of the Parallel Ice Sheet Model shows surge-type behavior. However, it then needs to be made very clear, where previous results were (not) confirmed and where new results are obtained. In contrast to the Author's claim, Van Pelt and Oerlemans also used a model version that included modeled pore water as enthalpy (page 348, second paragraph). The friction laws in the two publications look very similar to me. Please point out where the "simpler friction law' was improved. The visible main differences between the publications are the investigation of a water-terminating glacier instead of a land-terminating one (this has no obvious effect on the surge cycle as drawn in Figure 2), the parameters that were varied, and slight changes in the treatment of basal water.

The manuscript left me a bit concerned regarding the appreciation of the authors for previous work, and a tendency towards over-selling their own results. This begins in the title where Heinrich events are mentioned right at the start, while the manuscript deals with cyclic surging in a pretty generic ocean-terminating glacier, that is not obviously set up to resemble the situation in the Hudson Bay / Hudson Strait area, the source region of Heinrich events. I would suggest cutting the title to *The grow-and-surge instability in the Parallel Ice Sheet model* or something similar. While continuing through the manuscript, I would have enjoyed seeing more references to previous publications on glacier surging, and the similarities and differences in the underlying theory and results. This has already substantially improved since the first version of the manuscript, but I think it can be improved further. One example is the introduction of Figure 2 in Section 3.1, where a comparison of the model presented here to the binge-purge cycles of MacAyeal 1993 (and subsequent publications, e.ġ, Roberts et al, 2016) could make it easier for the reader to appreciate the additions presented in this manuscript. There also is a substantial body of research on growth-and-surge cycles in the context of mountain glaciers that could be referenced.

The newly introduced comparison of different values for the basal sliding exponent $q$ suffers from an ill-chosen reference velocity $u_c = 1\,\mathrm{m\,s^{-1}}$. This is the velocity for which the basal friction $\tau_b$ is independent of the exponent $q$. For comparing different flow law exponents, this value has to be in the range of typical sliding velocities. Using numbers obtained from the surge phase in Figure 3, the basal shears stress for ice sliding at $u_b = 1000$ m a$^{-1}$, and a $\tau_c$ value of $300\,\mathrm{kPa}$ varies over more than six orders of magnitude for the different values of q. It is highest for the plastic flow law with $\tau_b = 300\,\mathrm{kPa}$ (q=0) and decreases via $\tau_b = 9500\,\mathrm{Pa}$ (q=$\frac{1}{3}$ ) to $\tau_b = 9.5\,\mathrm{Pa}$ (q=1). For lower velocities relevant for surge initiation, this contrast is even more extreme. Van Pelt and Oerlemans chose $u_c = 100\,\mathrm{m\,a^{-1}}$, a much more realistic value. This might explain some of the differences in the results of the two studies.

I am not convinced by averaging the values for Figs. 3, 6-9, A1, considering that at least two thirds of the grounded domain are not affected by the surge at all. I would probably prefer one or two point measurements in the center line, or a line-average of this line.

---

## Author Response (AR2)

**Detailed response to the Editor on manuscript tc-2016-235**

"From Heinrich Events to cyclic ice streaming: the grow-and-surge instability in the Parallel Ice Sheet Model"

by J. Feldmann and A. Levermann

Dear Prof. Vieli,

We would like to thank you for the careful handling of the review process of our manuscript. We are glad that the reviewer who was already reviewing in the first revision round is happy with our changes and additions and accepted the revised manuscript as it is. At the same time we appreciate the important points raised by the second new reviewer. Addressing the comments and suggestions made by the Reviewer and the Editor, we think that they really add to our manuscript. Consequently, we hope that with our newly revised version of the manuscript we meet all the requirements by the Reviewer and the Editor. Following their suggestions, we extended the introduction, the methods section as well as the discussion/conclusion section, in particular referring and comparing to the study of vanPelt and Oerlemans (2012) who used an earlier version of the model to simulate a qualitatively different type of ice body. Two new figures have been added to the manuscript (Figs. A1 and A3).

Please find below the *reviewers' comments in italics* and our detailed response in blue. We have further attached a revised manuscript that highlights the changes in the submission, as well as a clean revised version.

Best wishes,
J. Feldmann and A. Levermann

**Editor Decision: Reconsider after major revisions (08 Jun 2017) by Andreas Vieli**

**Comments to the Author:**

*Dear Authors,*

*After 2 in general positive reviews of the first version of the manuscript and with some rather substantial suggestions for changes/revisions of one of the reviewers, the authors undertook major revisions and added additional experiments and modelling investigations and addressed the more technical points as well. As the revisions were rather substantial the paper has been sent to 2 further re-reviews, of which one of the reviewers had already reviewed the first version of the manuscript and was happy with the revised version of the manuscript. The second new reviewer had however some additional in my view valid comments that should be addressed before publication and which I outline below (the detailed review is given further below). The main points to be addressed are in brief:*

*1. to clarify the difference in the model and results of this study in relation to an earlier study by VanPelt and Oerlemans (2012) who used almost the same model. This should include*
*-a clarification in the model difference*
*-some better discussion/integration of the own and maybe new findings against/with Pelt and Oerlemans (2012)*
*-perhaps a clearer discussion of what the implications of 'ocean-terminating' really changes /means (compared to the land-based VanPelt Oerlemans (2012) study).*

We agree with the Editor that a better highlighting of the differences between the models and the applied experimental setups is of great help in order to clarify the difference between the two studies. Accordingly, we added a paragraph in the introduction referring to vanPelt and Oerlemans (2012) who also used PISM, briefly mentioning the main differences (p. 2, ll. 31-35). We detail these differences in the model section (p. 4, ll. 2-5, ll. 8-9) and the experimental setup section (p. 5, ll. 1-9). The main differences that we carve out lie in the friction law (linear vs. non-linear friction coefficient) and in the model setup (land-terminating glacier vs. marine ice-sheet-shelf system), besides other differences (spatially uniform vs. non-uniform surface mass balance and surface temperature, horizontal diffusion of basal melt water).
In this respect, we also revised the discussion/conclusion section, adding a more detailed comparison between our results and those from vanPelt and Oerlemans (2012), discussing the influence of the differences in the model/setup mentioned above on the results in three new paragraphs (p. 10, l. 16 – p.11, l. 6). This also includes the implications from the qualitatively different experimental setups, i.e., land-based glacier with unconfined tongue-shaped terminus vs. ocean-terminating ice-sheet-shelf system with the implication of non-buttressed vs. buttressed ice flow (p. 10, ll. 23-28).

*2. Related, this study should be embedded slightly better into the surge-modelling literature in the*

*introduction and discussion and therefore VanPelt and Oerlemans should likely be mentioned in the introduction, as their study is very close (with the difference of being land-based).*

We agree that this study was mentioned quite late in our previous version of the manuscript and that it makes a lot of sense to refer to it much earlier in the text. Following the suggestion by the Editor, now we also mention it in the introduction in the list of similar surge-related studies (p. 2, l. 11), as well as in a separate paragraph (p. 2, ll. 31-34, see our comment above). Furthermore, as suggested by the Reviewer, we included a couple of references to surging mountain glaciers, extending our introduction which only considered cyclic ice streaming and large-scale ice-sheet surging before (p. 1, ll. 15-18).

*3. link to HE-events (overselling): the used geometry seems pretty generic and perhaps together with the results not too close to the real HE-case (the modelled cycle-periods also do not match and in reality there was likely a slight retrograde slope into Hudson bay which would change things quite a bit regarding groundingline motion and feedbacks). Thus although a motivation by the HE-events seems ok, maybe the title could be rethought and be less strongly linked to HE-events (HE-event 'like', 'towards' HE-event cycles….???). At the end it is the choice of the authors of how to label the manuscript but i think one should be careful to also deliver to the raised expectations. Note, in the context of HE-event mechanisms and for completeness, in the introduction perhaps the recent literature of Alvarez-Solas (2013) and Bassis et al (2017) (see detailed references further below) are may be relevant (although not directly related to the mechanism investigated here, so not crucial).*

We see the Editor's point and at the same time are glad that he understands our idea of using Heinrich Events as a motivation. Following his suggestion we changed the title of the manuscript to "From cyclic ice streaming towards Heinrich-like events: the grow-and-surge instability in the Parallel Ice Sheet Model" in order to have a weaker link to Heinrich Events while at the same time keeping the motivation via Heinrich Events. We are glad to include the Editor's references to further studies that investigate externally-forced periodic surging and also added another study that proposed that Heinrich Events might not be triggered by purely ice intrinsic mechanisms (p. 2, ll. 3-5).

*4. the choice and implications of a rather low reference velocity (1m/y) should be addressed/ or clarified in relation to the variable flow law exponent 'q' (and related issues, see reviewer comment)*

Following the advice of the Reviewer, we included a paragraph discussing the choice of the parameter value, its meaning for the q-phi parameter study and also its implications regarding the differences in results between the two PISM studies (p. 10, l. 34 – p. 11, l. 5).

*5. I agree with the reviewer that the 'averaged grounded' visualization is not that ideal, and makes*

*it also difficult to compare to other studies (e.g. VanPelt and Oerlemans 2012). At least for some of the experiments maybe a average or max. centreline value could also be shown (as additional dashed lines or in appendix).*

We also agree that it makes a lot of sense to give an alternative to the averaged values. For this purpose we generated two figures that show the centerline average as well as two point values on this centerline (as also suggested by the Reviewer), analogous to Figs. 3 and A1 (now A2), respectively. As suggested by the Editor, we added these Figures to the Appendix (Figs. A1 and A3)

*For details see reviewer comment below*

*Editor Andreas Vieli*

**Reviewer comment to be addressed:**

*In their manuscript From Heinrich events to cyclic ice streaming, the grow-and- surge instability in the Parallel Ice Sheet model, Johannes Feldmann and Anders Levermann nicely describe their findings from surge-type experiments with the Parallel Ice Sheet Model.*
*In this respect, this manuscript is very similar to the Van Pelt and Oerlemans 2012 publication. There is nothing fundamentally wrong with this. Testing previously obtained results is an important process in science, and it is interesting to see that and under which parameters the current version of the Parallel Ice Sheet Model shows surge-type behavior.*

We would like to thank the Reviewer for taking the responsibility of reviewing our manuscript, the helpful advice and constructive criticism. We think that the Reviewer's valuable comments and suggestions, which we address below, were most useful in improving our manuscript.

*However, it then needs to be made very clear, where previous results were (not) confrmed and where new results are obtained. In contrast to the Author's claim, Van Pelt and Oerlemans also used a model version that included modeled pore water as enthalpy (page 348, second paragraph). The friction laws in the two publications look very similar to me. Please point out where the 'simpler friction law` was improved. The visible main differences between the publications are the investigation of a water-terminating glacier instead of a land- terminating one (this has no obvious effect on the surge cycle as drawn in Figure 2), the parameters that were varied, and slight changes in the treatment of basal water.*

We agree with the Reviewer that the differences between the models and also the obtained results were not made very clear in the previous version of our manuscript. Thus in the revised version we added a paragraph to the introduction, briefly mentioning the main differences between the models and experiments (p. 2, ll. 31-35). Extending the methods section, now we detail the differences in the friction law (linearly vs. non-linearly dependent on basal till water [p. 4, ll. 2-5] and the treatment of the basal till water [p. 4, ll. 8-9]). We also describe the differences in the experimental setup (boundary conditions and in particular the bed topography) in more detail and its consequences for the resulting qualitatively different types of modeled ice bodies (p. 5, ll. 1-9). Revising the discussion section, we added three paragraphs that discuss the main differences between 1) the models, 2) the experimental setups and 3) the velocity scaling parameter value and their effect on the results (p. 10, l. 16 – p.11, l. 6). We removed our statement on the enthalpy scheme.

*The manuscript left me a bit concerned regarding the appreciation of the authors for previous work, and a tendency towards over-selling their own results. This begins in the title where Heinrich events are mentioned right at the start, while the manuscript deals with cyclic surging in a pretty generic ocean-terminating glacier, that is not obviously set up to resemble the situation in duthe Hudson Bay / Hudson Strait area, the source region of Heinrich events. I would suggest cutting the*

*title to The grow-and-surge instability in the Parallel Ice Sheet model or something similar.*

We understand the the Reviewer's concern that the title might promise too much since our study is indeed not designed to reproduce Heinrich Events. Our idea of including Heinrich Events in the title was to use it as a motivation and also to reach a broader community. In order to weaken the link to Heinrich Events we followed the suggestion by the Editor and changed the title to: "From cyclic ice streaming towards Heinrich-like events: the grow-and-surge instability in the Parallel Ice Sheet Model".

*While continuing through the manuscript, I would have enjoyed seeing more references to previous publications on glacier surging, and the similarities and differences in the underlying theory and results. This has already substantially improved since the first version of the manuscript, but I think it can be improved further. One example is the introduction of Figure 2 in Section 3.1, where a comparison of the model presented here to the binge-purge cycles of MacAyeal 1993 (and subsequent publications, e.g, Roberts et al, 2016) could make it easier for the reader to appreciate the additions presented in this manuscript. There also is a substantial body of research on growth-and-surge cycles in the context of mountain glaciers that could be referenced.*

Following the Reviewer's suggestion we extended the first paragraph in Section 3.1, setting our study into context of previous work (p. 5, ll. 18-28). In this respect, we would also like to point to the introduction where we give a brief overview on previous surge-related work and also highlight the additions made by our study (p. 2, ll. 3-30). Regarding the literature on surging mountain glaciers we were glad to add several references to the introduction (p. 1, ll. 15-18) to also mention this type of surging (besides large-scale ice-sheet surging and cyclic ice streaming).

*The newly introduced comparison of different values for the basal sliding exponent q suffers from an ill-chosen reference velocity uc = 1m/s. This is the velocity for which the basal friction _b is independent of the exponent q. For comparing different flow law exponents, this value has to be in the range of typical sliding velocities. Using numbers obtained from the surge phase in Figure 3, the basal shears stress for ice sliding at ub = 1000 m/a, and a _c value of 300 kPa varies over more than six orders of magnitude for the different values of q. It is highest for the plastic flow law with _b = 300 kPa (q=0) and decreases via _b = 9500Pa (q=1/3 ) to _b = 9:5Pa (q=1). For lower velocities relevant for surge initiation, this contrast is even more extreme. Van Pelt and Oerlemans chose uc = 100m/a, a much more realistic value. This might explain some of the differences in the results of the two studies.*

We would like to thank the Reviewer for pointing to this important issue. Consequently, we included a paragraph in the discussion/conclusion section, discussing the implications of our choice of the velocity scaling parameter on the q-phi parameter study and on the comparability of the results mentioned by the Reviewer (p. 10, l. 34 – p. 11, l. 5).

*I am not convinced by averaging the values for Figs. 3, 6-9, A1, considering that at least two thirds of the grounded domain are not affected by the surge at all. I would probably prefer one or two point measurements in the center line, or a line-average of this line.*

We agree with the Reviewer that values along the centerline give deeper insight into the surge dynamics. Following the Reviewer's suggestion now we also show curves for the centerline-average as well as two point measurements on the centerline (Figs. A1 and A3).

*References:*

[revised manuscript text omitted]

---

## Author Response (AR3)

**Detailed response to the Editor on manuscript tc-2016-235**

"From cyclic ice streaming towards Heinrich-like events: the grow-and-surge instability in the Parallel Ice Sheet Model"

by J. Feldmann and A. Levermann

Dear Prof. Vieli,

We would like to thank you for your fast response and are glad to hear that our manuscript is now only subject to minor corrections before publication. We are thankful for the technical points raised by the Editor and address all of them.

Please find below the *Editor's comments in italics* and our detailed response in blue. We have further attached a revised manuscript that highlights the changes in the submission, as well as a clean revised version.

Best wishes,
J. Feldmann and A. Levermann

**Comments to the Author:**
**Final decision of editor based on comments of two initial reviews, major revisions undertaken and two re-reviews of the revised version.**

*Dear J. Feldmann,*
*After substantial revisions in response of two initially critical but overall very positive reviews, this paper has been reviewed again (one referee of first round, one new referee), with some substantial suggestion for revisions of the new referee.*
*These critical points were addressed mostly very well and in detail in the 2nd revision of the manuscript and I thank the authors for their carful revisions and detailed response.*
*After the viewing the revised version and considering all the information from the response and reviews the manuscript seems now very close to final publication in TC.*
*However, before official acceptance there are a few technical details that need correcting (see list below) and one remaining point flagged in the last review that could in my view be addressed a bit better (left also to author judgment).*

*This last point refers to the reference to the earlier somewhat similar study by Pelt and Oerlemans (2012). In general the revised version makes the links and differences to this study much clearer (method, model, results), but personally I think it would be useful to know before the results and in the introduction what the main findings/conclusions of the vanPelt study were, as this study is so close. Currently, this is done in the final discussion, but I would personally prefer, to very briefly outline their for tis study relevant results in the introduction (besides the difference in model/methods), maybe in the last paragraph p. 2 lines 31-35.*
*I leave the final decision on this to the authors, but I think knowing upfront what others found/concluded would be useful here.*

We are thankful for the Editor's advice and followed his suggestion, adding a brief summary of the findings in van Pelt and Oerlemans (2012) most relevant to our study at the end of the introduction.

*Detailed more technical points to correct/address:*

*p. 2 line 28: 'sentence with '…visualized in a novel way..' , I do not quite know which graphs/figures/visualization are referred to here and what should really be novel here. Is it the conceptual feedback loops? But seems not very 'visually novel' to me but rather conceptual and at least parts of it seemed familiar to me. I would simply leave this sentence away (to avoid issue of what is really 'novel') or at least rephrase clearer.*

Done (rephrased).

*p. 2 line 34: I would be more specific and integrate 'marine terminating' or similar (instead of just*

*'bed topography, boundary conditions')*

Done.

*p. 7 line 15 and Fig 7: maybe I am picky but the frequency first increases from 10% to 30% and then decreases.*

Corrected.

*p. 10 line 16: I would be more explicit here to which study you compare here: e.g. something like'… differences in results compared to the PISM study of Van Pelt and Oerlemans (2012) are a …'*

Rephrased as suggested.

*p. 10 line 23/24: 'qualitatively different ICE MASSES' seems a bit awkward, you probably mean 'qualitatively different ice sheet GEOMTERIES or CHARACTERISTICS'?*

Rephrased.

*p. 10 line 28/29: formulation is here just appears somewhat confusing (contradicting): same order of magnitude but 5 times faster; I guess you similar but still substantially different. Maybe reformulate differently.*

Rephrased.

*p. 11 paragraph lines 7-15: could add how you results compares to van pelt and oerlemans (2012). In particular the found decreasing amplitude with decreasing accumulation contrasting the result of to other studies (p. 10-11 line 11), is as far as I know consistent at least with the van Pelt (although they did not exactly vary accum. But ELA). Would probably strengthen your case.*

Thanks for the hint! Included.

*Figure 8 caption: I think the caption on (a) and (b) is here the wrong way round: (a) should be the from 10degree equilibrium to 8 degree, and (b) the increase from 8 degree to …*
*Currently is the other way round!?*

Corrected.

*Figure A1 caption : maybe one could add at end of caption '… instead of the area grounded averaged values.'*

Added.

*Figure A2 caption: 'Additional timeseries to results in Fig. 3 of ….'*

Added.

*Figure A3 caption: should it not refer to Fig. A2: '…complementing Fig. A2…' as it shows the same quantities as in fig A2 but for centreline values. Or maybe refer to both 'Fig A2 and 3'.*

It should indeed be this way and was displayed incorrectly only in the manuscript version with highlighted changes.

*Andreas Vieli (editor), 13th July 2017*

[revised manuscript text omitted]